# Laplace Redux – Effortless Bayesian Deep Learning

**Erik Daxberger**[*,c,m]     **Agustinus Kristiadi**[*,t]     **Alexander Immer**[*,e,p]     **Runa Eschenhagen**[*,t]
**Matthias Bauer**[d]     **Philipp Hennig**[t,m]

[c]University of Cambridge
[m]MPI for Intelligent Systems, Tübingen
[t]University of Tübingen
[e]Department of Computer Science, ETH Zurich
[p]Max Planck ETH Center for Learning Systems
[d]DeepMind, London

## Abstract

Bayesian formulations of deep learning have been shown to have compelling theoretical properties and offer practical functional benefits, such as improved predictive uncertainty quantification and model selection. The Laplace approximation (LA) is a classic, and arguably the simplest family of approximations for the intractable posteriors of deep neural networks. Yet, despite its simplicity, the LA is not as popular as alternatives like variational Bayes or deep ensembles. This may be due to assumptions that the LA is expensive due to the involved Hessian computation, that it is difficult to implement, or that it yields inferior results. In this work we show that these are misconceptions: we (i) review the range of variants of the LA including versions with minimal cost overhead; (ii) introduce `laplace`, an easy-to-use software library for PyTorch offering user-friendly access to all major flavors of the LA; and (iii) demonstrate through extensive experiments that the LA is competitive with more popular alternatives in terms of performance, while excelling in terms of computational cost. We hope that this work will serve as a catalyst to a wider adoption of the LA in practical deep learning, including in domains where Bayesian approaches are not typically considered at the moment.

`laplace` **library:** https://github.com/AlexImmer/Laplace
**Experiments:** https://github.com/runame/laplace-redux

## 1 Introduction

Despite their successes, modern neural networks (NNs) still suffer from several shortcomings that limit their applicability in some settings. These include (i) poor calibration and overconfidence, especially when the data distribution shifts between training and testing [1], (ii) catastrophic forgetting of previously learned tasks when continuously trained on new tasks [2], and (iii) the difficulty of selecting suitable NN architectures and hyperparameters [3]. Bayesian modeling [4, 5] provides a principled and unified approach to tackle these issues by (i) equipping models with robust uncertainty estimates [6], (ii) enabling models to learn continually by capturing past information [7], and (iii) allowing for automated model selection by optimally trading off data fit and model complexity [8].

Even though this provides compelling motivation for using *Bayesian neural networks* (BNNs) [9], they have not gained much traction in practice. Common criticisms include that BNNs are difficult

---

[*]Equal contributors; author ordering sampled uniformly at random. Correspondence to: ead54@cam.ac.uk, agustinus.kristiadi@uni-tuebingen.de, alexander.immer@inf.ethz.ch, runa.eschenhagen@student.uni-tuebingen.de.

35th Conference on Neural Information Processing Systems (NeurIPS 2021).

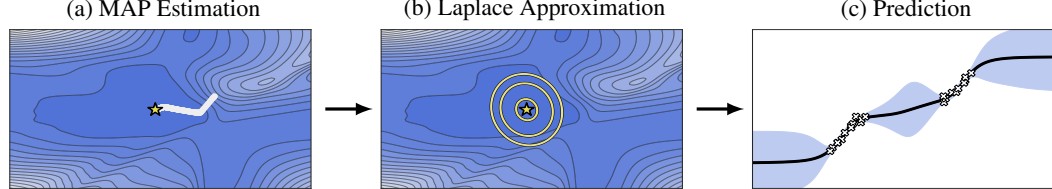

| (a) MAP Estimation | (b) Laplace Approximation | (c) Prediction |

**Figure 1: Probabilistic predictions with the Laplace approximation in three steps. (a)** We find a MAP estimate (yellow star) via standard training (background contours = log-posterior landscape on the two-dimensional PCA subspace of the SGD trajectory [30]). **(b)** We locally approximate the posterior landscape by fitting a Gaussian centered at the MAP estimate (yellow contours), with covariance matrix equal to the negative inverse Hessian of the loss at the MAP—this is the Laplace approximation (LA). **(c)** We use the LA to make predictions with *predictive uncertainty estimates*— here, the black curve is the predictive mean, and the shading covers the 95% confidence interval.

to implement, finicky to tune, expensive to train, and hard to scale to modern models and datasets. For instance, popular variational Bayesian methods [10–12, etc.] require considerable changes to the training procedure and model architecture. Also, their optimization process is slower and typically more unstable unless carefully tuned [13]. Other methods, such as deep ensembles [14], Monte Carlo dropout [6], and SWAG [15] promise to bring uncertainty quantification to standard NNs in simple manners. But these methods either require a significant cost increase compared to a single network, have limited empirical performance, or an unsatisfying Bayesian interpretation.

In this paper, we argue that the Laplace approximation (LA) is a simple and cost-efficient, yet competitive approximation method for inference in Bayesian deep learning. First proposed in this context by MacKay [16], the LA dates back to the 18th century [17]. It locally approximates the posterior with a Gaussian distribution centered at a local maximum, with covariance matrix corresponding to the local curvature. Two key advantages of the LA are that the local maximum is readily available from standard *maximum a posteriori* (MAP) training of NNs, and that curvature estimates can be easily and efficiently obtained thanks to recent advances in second-order optimization, both in terms of more efficient approximations to the Hessian [18–20] and easy-to-use software libraries [21]. Together, they make the LA practical and readily applicable to many already-trained NNs—the LA essentially enables practitioners to turn their high-performing point-estimate NNs into BNNs easily and quickly, without loss of predictive performance. Furthermore, the LA to the marginal likelihood may even be used for Bayesian model selection or NN training [8, 22]. Figure 1 provides an intuition of the LA—we first fit a point estimate of the model and then estimate a Gaussian distribution around that.

Yet, despite recent progress in scaling and improving the LA for deep learning [23–29], it is far less widespread than other methods. This is likely due to misconceptions, like that the LA is hard to implement due to the Hessian computation, that it must necessarily perform worse than the competitors due to its local nature, or quite simply that it is old and too simple. Here, we show that these are indeed misconceptions. Moreover, we argue that the LA deserves a wider adoption in both practical and research-oriented deep learning. To this end, our work makes the following contributions:

1. We first survey recent advances and present the key components of scalable and practical Laplace approximations in deep learning (Section 2).
2. We then introduce `laplace`, an easy-to-use PyTorch-based library for "turning a NN into a BNN" via the LA (Section 3). `laplace` implements a wide range of different LA variants.
3. Lastly, using `laplace`, we show in an extensive empirical study that the LA is competitive to alternative approaches, especially considering how simple and cheap it is (Section 4).

## 2 The Laplace Approximation in Deep Learning

The LA can be used in two different ways to benefit deep learning: Firstly, we can use the LA to approximate the model's *posterior distribution* (see Eq. (5) below) to enable *probabilistic predictions* (as also illustrated in Fig. 1). Secondly, we can use the LA to approximate the *model evidence* (see Eq. (6)) to enable *model selection* (e.g. hyperparameter tuning).

The canonical form of (supervised) deep learning is that of empirical risk minimization. Given, e.g., an i.i.d. classification dataset $\mathcal{D} := \{(x_n \in \mathbb{R}^M, y_n \in \mathbb{R}^C)\}_{n=1}^N$, the weights $\theta \in \mathbb{R}^D$ of an $L$-layer NN $f_\theta : \mathbb{R}^M \to \mathbb{R}^C$ are trained to minimize the (regularized) empirical risk, which typically decomposes into a sum over empirical loss terms $\ell(x_n, y_n; \theta)$ and a regularizer $r(\theta)$,

$$\theta_{\text{MAP}} = \arg\min_{\theta \in \mathbb{R}^D} \mathcal{L}(\mathcal{D}; \theta) = \arg\min_{\theta \in \mathbb{R}^D} \left( r(\theta) + \sum_{n=1}^N \ell(x_n, y_n; \theta) \right). \quad (1)$$

From the Bayesian viewpoint, these terms can be identified with i.i.d. log-***likelihoods*** and a log-***prior***, respectively and, thus, $\theta_{\text{MAP}}$ is indeed a ***maximum a-posteriori (MAP)*** estimate:

$$\ell(x_n, y_n; \theta) = -\log p(y_n \mid f_\theta(x_n)) \qquad \text{and} \qquad r(\theta) = -\log p(\theta) \quad (2)$$

For example, the widely used weight regularizer $r(\theta) = \frac{1}{2}\gamma^{-2}\|\theta\|^2$ (a.k.a. weight decay) corresponds to a centered Gaussian prior $p(\theta) = \mathcal{N}(\theta; 0, \gamma^2 I)$, and the cross-entropy loss amounts to a categorical likelihood. Hence, the exponential of the negative training loss $\exp(-\mathcal{L}(\mathcal{D}; \theta))$ amounts to an ***unnormalized posterior***. By normalizing it, we obtain

$$p(\theta \mid \mathcal{D}) = \tfrac{1}{Z} p(\mathcal{D} \mid \theta) p(\theta) = \tfrac{1}{Z} \exp(-\mathcal{L}(\mathcal{D}; \theta)), \qquad Z := \int p(\mathcal{D} \mid \theta) p(\theta) \, d\theta \quad (3)$$

with an intractable ***normalizing constant*** $Z$. ***Laplace approximations*** [17] use a second-order expansion of $\mathcal{L}$ around $\theta_{\text{MAP}}$ to construct a Gaussian approximation to $p(\theta \mid \mathcal{D})$. I.e. we consider:

$$\mathcal{L}(\mathcal{D}; \theta) \approx \mathcal{L}(\mathcal{D}; \theta_{\text{MAP}}) + \tfrac{1}{2}(\theta - \theta_{\text{MAP}})^\intercal \left( \nabla_\theta^2 \mathcal{L}(\mathcal{D}; \theta)|_{\theta_{\text{MAP}}} \right) (\theta - \theta_{\text{MAP}}), \quad (4)$$

where the first-order term vanishes at $\theta_{\text{MAP}}$. Then we can identify the Laplace approximation as

— Laplace posterior approximation —

$$p(\theta \mid \mathcal{D}) \approx \mathcal{N}(\theta; \theta_{\text{MAP}}, \Sigma) \qquad \text{with} \qquad \Sigma := - \left( \nabla_\theta^2 \mathcal{L}(\mathcal{D}; \theta)|_{\theta_{\text{MAP}}} \right)^{-1}. \quad (5)$$

The normalizing constant $Z$ (which is typically referred to as the *marginal likelihood* or *evidence*) is useful for model selection and can also be approximated as

— Laplace approximation of the evidence —

$$Z \approx \exp(-\mathcal{L}(\mathcal{D}; \theta_{\text{MAP}})) (2\pi)^{D/2} (\det \Sigma)^{1/2}. \quad (6)$$

See Appendix A for more details. Thus, to obtain the approximate posterior, we first need to find the argmax $\theta_{\text{MAP}}$ of the log-posterior function, i.e. do "standard" deep learning with regularized empirical risk minimization. The only *additional* step is to compute the inverse of the Hessian matrix at $\theta_{\text{MAP}}$ (see Figure 1(b)). The LA can therefore be constructed *post-hoc* to a pre-trained network, even one downloaded off-the-shelf. As we discuss below, the Hessian computation can be offloaded to recently advanced automatic differentiation libraries [21]. LAs are widely used to approximate the posterior distribution in logistic regression [31], Gaussian process classification [32, 33], and also for Bayesian neural networks (BNNs), both shallow [34] and deep [23]. The latter is the focus of this work.

Generally, any prior with twice differentiable log-density can be used. Due to the popularity of the weight decay regularizer, we assume that the prior is a zero-mean Gaussian $p(\theta) = \mathcal{N}(\theta; 0, \gamma^2 I)$ unless stated otherwise.[2] The Hessian $\nabla_\theta^2 \mathcal{L}(\mathcal{D}; \theta)|_{\theta_{\text{MAP}}}$ then depends both on the (simple) log-prior / regularizer and the (complicated) log-likelihood / empirical risk:

$$\nabla_\theta^2 \mathcal{L}(\mathcal{D}; \theta)|_{\theta_{\text{MAP}}} = -\gamma^{-2} I - \sum_{n=1}^N \nabla_\theta^2 \log p(y_n \mid f_\theta(x_n))|_{\theta_{\text{MAP}}}. \quad (7)$$

A naive implementation of the Hessian is infeasible because the second term in Eq. (7) scales quadratically with the number of network parameters, which can be in the millions or even billions [35, 36]. In recent years, several works have addressed scalability, as well as other factors that affect approximation quality and predictive performance of the LA. In the following, we identify, review, and discuss four key components that allow LAs to scale and perform well on modern deep architectures. See Fig. 2 for an overview and Appendix B for a more detailed version of the review and discussion.

**Four Components of Scalable Laplace Approximations for Deep Neural Networks**

### ① Inference over all Weights or Subsets of Weights

In most cases, it is possible to treat *all* weights probabilistically when using appropriate approximations of the Hessian, as we discuss below in ②. Another simple way to scale the LA to large NNs

---

[2]One can also consider a per-layer or even per-parameter weight decay, which corresponds to a more general, but still comparably simple Gaussian prior. In particular, the Hessian of this prior is still diagonal and constant.

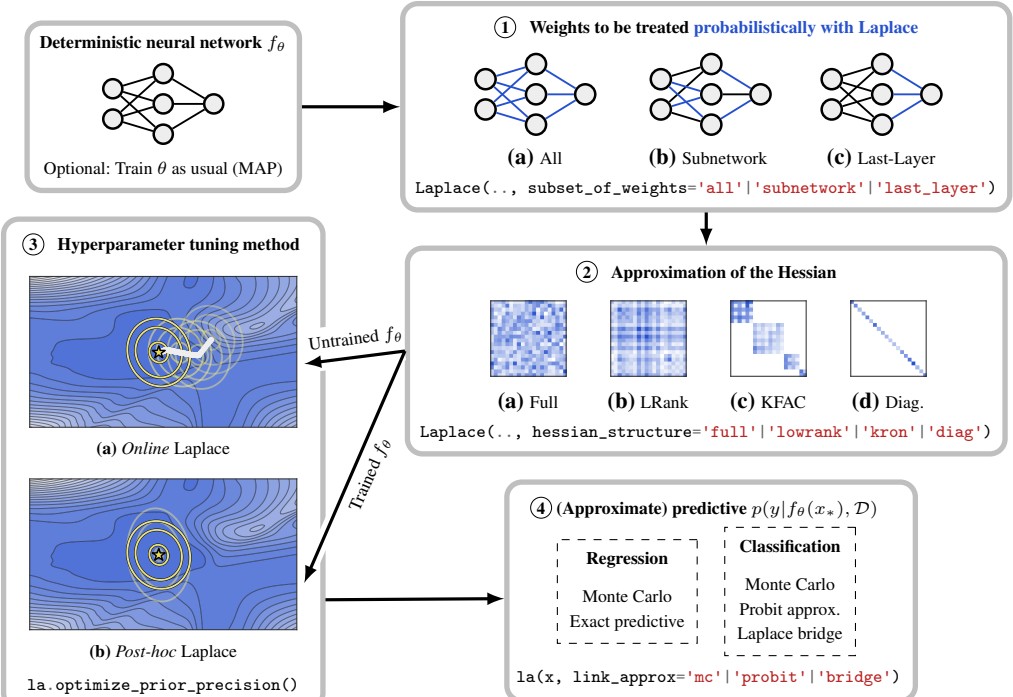

**Figure 2:** Four key components to scale and apply the LA to a neural network $f_\theta$ (with randomly-initialized or pre-trained weights $\theta$), with corresponding `laplace` code. ① We first choose which part of the model we want to perform inference over with the LA. ② We then select how to to approximate the Hessian. ③ We can then perform model selection using the evidence: **(a)** If we started with an untrained model $f_\theta$, we can jointly train the model and use the evidence to tune hyperparameters *online*. **(b)** If we started with a pre-trained model, we can use the evidence to tune the hyperparameters *post-hoc*. Here, shades represent the loss landscape, while contours represent LA log-posteriors—faded contours represent intermediate iterates during hyperparameter tuning to obtain the final log-posterior (thick yellow contours). ④ Finally, to make predictions for a new input $x_*$, we have several options for computing/approximating the predictive distribution $p(y|f_\theta(x_*), \mathcal{D})$.

(without Hessian approximations) is the ***subnetwork LA*** [27], which only treats a *subset* of the model parameters probabilistically with the LA and leaves the remaining parameters at their MAP-estimated values. An important special case of this applies the LA to only the *last linear layer* of an $L$-layer NN, while fixing the feature extractor defined by the first $L - 1$ layers at its MAP estimate [37, 28]. This ***last-layer LA*** is cost-effective yet compelling both theoretically and in practice [28].

② **Hessian Approximations and Their Factorizations**

One advance in second-order optimization that the LA can benefit from are positive semi-definite approximations to the (potentially indefinite) Hessian of the log-likelihoods of NNs in the second term of Eq. (7) [38]. The ***Fisher information matrix*** [39], abbreviated as *the Fisher* and defined by

$$F := \sum_{n=1}^{N} \mathbb{E}_{\widehat{y} \sim p(y \,|\, f_\theta(x_n))} \left[ (\nabla_\theta \log p(\widehat{y} \,|\, f_\theta(x_n))|_{\theta_{\mathrm{MAP}}})(\nabla_\theta \log p(\widehat{y} \,|\, f_\theta(x_n))|_{\theta_{\mathrm{MAP}}})^\intercal \right], \quad (8)$$

is one such choice.[3] One can also use the ***generalized Gauss-Newton matrix (GGN)*** matrix [41]

$$G := \sum_{n=1}^{N} J(x_n) \left( \nabla_f^2 \log p(y_n \,|\, f)|_{f=f_{\theta_{\mathrm{MAP}}}(x_n)} \right) J(x_n)^\intercal, \quad (9)$$

where $J(x_n) := \nabla_\theta f_\theta(x_n)|_{\theta_{\mathrm{MAP}}}$ is the NN's Jacobian matrix. As the Fisher and GGN are equivalent for common log-likelihoods [38], we will henceforth refer to them interchangeably. In deep LAs, they have emerged as the default choice [23, 24, 28, 29, 27, 26, etc.].

---

[3]If, instead of taking expectation in (8), we use the training label $y_n$, we call the matrix the ***empirical Fisher***, which is distinct from the Fisher [38, 40].

As $F$ and $G$ are still quadratically large, we typically need further factorization assumptions. The most lightweight is a ***diagonal factorization*** which ignores off-diagonal elements [42, 43]. More expressive alternatives are block-diagonal factorizations such as ***Kronecker-factored approximate curvature (KFAC)*** [18–20], which factorizes each within-layer Fisher[4] as a Kronecker product of two smaller matrices. KFAC has been successfully applied to the LA [23, 24] and can be improved by low-rank approximations of the KFAC factors [29] by leveraging their eigendecompositions [44]. Finally, recent work has studied/enabled ***low-rank approximations*** of the Hessian/Fisher [45–47].

### ③ Hyperparameter Tuning

As with all approximate inference methods, the performance of the LA depends on the (hyper)parameters of the prior and likelihood. For instance, it is typically beneficial to tune the prior variance $\gamma^2$ used for inference [23, 28, 27, 26, 22]. Commonly, this is done through ***cross-validation***, e.g. by maximizing the validation log-likelihood [23, 48] or, additionally, using out-of-distribution data [28, 49]. When using the LA, however, ***marginal likelihood maximization*** (a.k.a. ***empirical Bayes*** or ***the evidence framework*** [34, 50]) constitutes a more principled alternative to tune these hyperparameters, and requires no validation data. Immer et al. [22] showed that marginal likelihood maximization with LA can work in deep learning and even be performed in an online manner jointly with the MAP estimation. Note that such approach is not necessarily feasible for other approximate inference methods because most do not provide an estimate of the marginal likelihood. Other recent approaches for hyperparameter tuning for the LA include Bayesian optimization [51] or the addition of dedicated, trainable hidden units for the sole purpose of uncertainty tuning [49].

### ④ Approximate Predictive Distribution

To predict using a posterior (approximation) $p(\theta \,|\, \mathcal{D})$, we need to compute $p(y \,|\, f(x_*), \mathcal{D}) = \int p(y \,|\, f_\theta(x_*)) \, p(\theta \,|\, \mathcal{D}) \, d\theta$ for any test point $x_* \in \mathbb{R}^n$, which is intractable in general. The simplest but most general approximation to $p(y \,|\, x_*, \mathcal{D})$ is Monte Carlo integration using $S$ samples $(\theta_s)_{s=1}^S$ from $p(\theta \,|\, \mathcal{D})$: $p(y \,|\, f(x_*), \mathcal{D}) \approx S^{-1} \sum_{s=1}^S p(y \,|\, f_{\theta_s}(x_*))$. However, for LAs with GGN and Fisher Hessian approximations Monte Carlo integration can perform poorly [48, 26]. Immer et al. [26] attribute this to the inconsistency between Hessian approximation and the predictive and suggest to use a linearized predictive instead, which can also be useful for theoretic analyses [28]. For the last-layer LA, the Hessian coincides with the GGN and the linearized predictive is exact.

The predictive of a ***linearized neural network*** with a LA approximation to the posterior $p(\theta \,|\, \mathcal{D}) \approx \mathcal{N}(\theta; \theta_{\mathrm{MAP}}, \Sigma)$ results in a Gaussian distribution on neural network outputs $f_* := f(x_*)$ and therefore enables simple approximations or even a closed-form solution. The distribution on the outputs is given by $p(f_* \,|\, x_*, \mathcal{D}) \approx \mathcal{N}(f_*; f_{\theta_{\mathrm{MAP}}}(x_*), J(x_*)^\intercal \Sigma J(x_*))$ and is typically significantly lower-dimensional (number of outputs $C$ instead of parameters $D$). It can also be inferred entirely in function space as a Gaussian process [25, 26]. Given the distribution on outputs $f_*$, the predictive distribution can be obtained by integration against the likelihood: $p(y \,|\, x_*, \mathcal{D}) = \int p(y \,|\, f_*) p(f_* \,|\, x_*, \mathcal{D}) \, d\theta$. In the case of regression with a Gaussian likelihood with variance $\sigma^2$, the solution can even be obtained analytically: $p(y \,|\, x_*, \mathcal{D}) \approx \mathcal{N}(y; f_{\theta_{\mathrm{MAP}}}(x_*), J(x_*)^\intercal \Sigma J(x_*) + \sigma^2 I)$. For non-Gaussian likelihoods, e.g. in classification, a further approximation is needed. Again, the simplest approximation to this is ***Monte Carlo integration***. In the binary case, we can employ the ***probit approximation*** [31, 16] which approximates the logistic function with the probit function. In the multi-class case, we can use its generalization, the ***extended probit approximation*** [52]. Finally, first proposed for non-BNN applications [53, 54], the ***Laplace bridge*** approximates the softmax-Gaussian integral via a Dirichlet distribution [55]. The key advantage is that it yields a *distribution* of the integral solutions.

## 3   `laplace`: A Toolkit for Deep Laplace Approximations

Implementing the LA is non-trivial, as it requires efficient computation and storage of the Hessian. While this is not fundamentally difficult, there exists no complete, easy-to-use, and standardized implementation of various LA flavors—instead, it is common for deep learning researchers to repeatedly re-implement the LA and Hessian computation with varying efficiency [56–58, etc.]. An efficient implementation typically requires hundreds of lines of code, making it hard to quickly prototype

---

[4]The elements $F$ or $G$ corresponding to the weight $W_l \subseteq \theta$ of the $l$-th layer of the network.

```
1  from laplace import Laplace
2
3  # Load pre-trained model
4  model = load_map_model()
5
6  # Define and fit LA variant with custom settings
7  la = Laplace(model, 'classification',
8                subset_of_weights='all',
9                hessian_structure='diag')
10 la.fit(train_loader)
11 la.optimize_prior_precision(method='CV',
12                              val_loader=val_loader)
13
14 # Make prediction with custom predictive approx.
15 pred = la(x, pred_type='glm', link_approx='probit')
```

**Listing 1:** Fit diagonal LA over all weights of a pre-trained classification model, do *post-hoc* tuning of the prior precision hyperparameter using cross-validation, and make a prediction for input $x$ with the probit approximation.

```
1  from laplace import Laplace
2
3  # Load un- or pre-trained model
4  model = load_map_model()
5
6  # Fit default, recommended LA variant:
7  # Last-layer KFAC LA
8  la = Laplace(model, 'regression')
9  la.fit(train_loader)
10
11 # Differentiate marginal likelihood w.r.t.
12 # prior precision and observation noise
13 ml = la.marglik(prior_precision=prior_prec,
14                 sigma_noise=obs_noise)
15 ml.backward()
```

**Listing 2:** Fit KFAC LA over the last layer of a pre- or un-trained regression model and differentiate its marginal likelihood w.r.t. some hyperparameters for *post-hoc* hyperparameter tuning or online empirical Bayes (see Immer et al. [22]).

with the LA. To address this, we introduce `laplace`: a simple, easy-to-use, extensible library for scalable LAs of deep NNs in PyTorch [59]. `laplace` enables *all* sensible combinations of the four components discussed in Section 2—see Fig. 2 for details. Listings 1 and 2 show code examples.

The core of `laplace` consists of efficient implementations of the LA's key quantities: (i) posterior (i.e. Hessian computation and storage), (ii) marginal likelihood, and (iii) posterior predictive. For (i), to take advantage of advances in automatic differentiation, we outsource the Hessian computation to state-of-the-art, optimized second-order optimization libraries: BackPACK [21] and ASDL [60]. Moreover, we design `laplace` in a modular manner that makes it easy to add new backends and approximations in the future. For (ii), we follow Immer et al. [22] in our implementation of the LA's marginal likelihood—it is thus both efficient and differentiable and allows the user to implement both *online* and *post-hoc* marginal likelihood tuning, cf. Listing 2. Note that `laplace` also supports standard cross-validation for hyperparameter tuning [23, 28], as shown in Listing 1. Finally, for (iii), `laplace` supports all approximations to the posterior predictive distribution discussed in Section 2—it thus provides the user with flexibility in making predictions, depending on the computational budget.

**Default behavior** To abstract away from a large number of options available (Section 2), we provide the following default choices based on our extensive experiments (Section 4); they should be applicable and perform decently in the majority of use cases: we assume a pre-trained network and treat only the last-layer weights probabilistically (last-layer LA), use the KFAC factorization of the GGN and tune the hyperparameters *post-hoc* using empirical Bayes. To make predictions, we use the closed-form Gaussian predictive distribution for regression and the (extended) probit approximation for classification. Of course, the user can pick custom choices (Listings 1 and 2).

**Limitations** Because `laplace` employs external libraries (BackPACK [21] and ASDL [60]) as backends, it inherits the available choices of Hessian factorizations from these libraries. For instance, the LA variant proposed by Lee et al. [29] can currently not be implemented via `laplace`, because neither backend supports eigenvalue-corrected KFAC [44] (yet).

## 4   Experiments

We benchmark various LAs implemented via `laplace`. Section 4.1 addresses the question of "which are the best design choices for the LA", in light of Figure 2. Section 4.2 shows that the LA is competitive to strong Bayesian baselines in in-distribution, dataset-shift, and out-of-distribution (OOD) settings. We then showcase some applications of the LA in downstream tasks. Section 4.3 demonstrates the applicability of the (last-layer) LA on various data modalities and NN architectures (including transformers [61])—settings where other Bayesian methods are challenging to implement. Section 4.4 shows how the LA can be used as an easy-to-use yet strong baseline in continual learning. In all results, arrows behind metric names denote if lower ($\downarrow$) or higher ($\uparrow$) values are better.

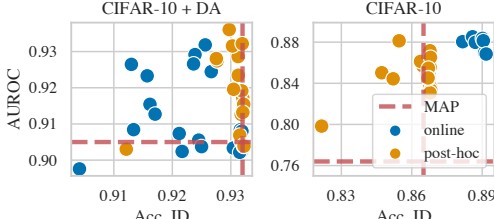

**Figure 3:** In- vs. out-of-distribution (ID and OOD, resp.) performance on CIFAR-10 of different LA configurations (dots), each being a combination of settings for 1) subset-of-weights, 2) covariance structure, 3) hyperparameter tuning, and 4) predictive approximation (see Appendix C.1 for details). "DA" stands for "data augmentation". Post-hoc performs better with DA and a strong pre-trained network, while online performs better without DA where optimal hyperparameters are unknown.

**Table 1:** OOD detection performance averaged over all test sets (see Appendix C.2 for details). Confidence is defined as the max. of the predictive probability vector [62] (e.g. Confidence([0.7, 0.2, 0.1]) = 0.7). LA and especially LA* reduce the overconfidence of MAP and achieve better results than the VB, CSGHMC (HMC), and SWAG (SWG) baselines.

| | Confidence ↓ | | AUROC ↑ | |
|---|---|---|---|---|
| **Methods** | **MNIST** | **CIFAR-10** | **MNIST** | **CIFAR-10** |
| MAP | 75.0±0.4 | 76.1±1.2 | 96.5±0.1 | 92.1±0.5 |
| DE | 65.7±0.3 | 65.4±0.4 | 97.5±0.0 | 94.0±0.1 |
| VB | 73.2±0.8 | 58.8±0.7 | 95.8±0.2 | 88.7±0.3 |
| HMC | 69.2±1.7 | 69.4±0.6 | 96.1±0.2 | 90.6±0.2 |
| SWG | 75.8±0.3 | 68.1±2.3 | 96.5±0.1 | 91.3±0.8 |
| LA | 67.5±0.4 | 69.0±1.3 | 96.2±0.2 | 92.2±0.5 |
| LA* | 56.1±0.5 | 55.7±1.2 | 96.4±0.2 | 92.4±0.5 |

### 4.1 Choosing the Right Laplace Approximation

In Section 2 we presented multiple options for each component of the design space of the LA, resulting in a large number of possible combinations, all of which are supported by `laplace`. Here, we try to reduce this complexity and make suggestions for sensible default choices that cover common application scenarios. To this end, we performed a comprehensive comparison between most variants; we measured in- and out-of-distribution performance on standard image classification benchmarks (MNIST, FashionMNIST, CIFAR-10) but also considered the computational complexity of each variant. We provide details of the comparison and a list of the considered variants in Appendix C.1 and summarize the main arguments and take-aways in the following.

**Hyperparameter tuning and parameter inference.** We can apply the LA purely *post-hoc* (only tune hyperparameters of a pre-trained network) or online (tune hyperparameters and train the network jointly, as e.g. suggested by Immer et al. [22]). We find that the online LA only works reliably when it is applied to all weights of the network. In contrast, applying the LA *post-hoc* only on the last layer instead of all weights typically yields better performance due to less underfitting, and is significantly cheaper. For problems where a pre-trained network or optimal hyperparameters are available, e.g. for well-studied data sets, we, therefore, suggest using the *post-hoc* variant on the last layer. This LA has the benefit that it has minimal overhead over a standard neural network forward pass (cf. Fig. 5) while performing on par or better than state-of-the-art approaches (cf. Fig. 4). When hyperparameters are unknown or no validation data is available, we suggest training the neural network online by optimizing the marginal likelihood, following Immer et al. [22] (cf Section 4.4). Figure 3 illustrates this on CIFAR-10: for CIFAR-10 with data augmentation, strong pre-trained networks and hyperparameters are available and the *post-hoc* methods directly profit from that while the online methods merely reach the same performance. On the less studied CIFAR-10 without data augmentation, the online method can improve the performance over the *post-hoc* methods.

**Covariance approximation and structure.** Generally, we find that a more expressive covariance approximation improves performance, as would be expected. However, a full covariance is in most cases intractable for full networks or networks with large last layers. The KFAC structured covariance provides a good trade-off between expressiveness and speed. Diagonal approximations perform significantly worse than KFAC and are therefore not suggested. Independent of the structure, we find that the empirical Fisher (EF) approximations perform better on out-of-distribution detection tasks while GGN approximations tend to perform better on in-distribution metrics.

**Predictive distribution.** Considering in- and out-of-distribution (OOD) performance as well as cost, the probit provides the best approximation to the predictive for the last-layer LA. MC integration can sometimes be superior for OOD detection but at an increased computational cost. The Laplace bridge has the same cost as the probit approximation but typically provides inferior results in our experiments. When using the LA online to optimize hyperparameters, we find that the resulting MAP

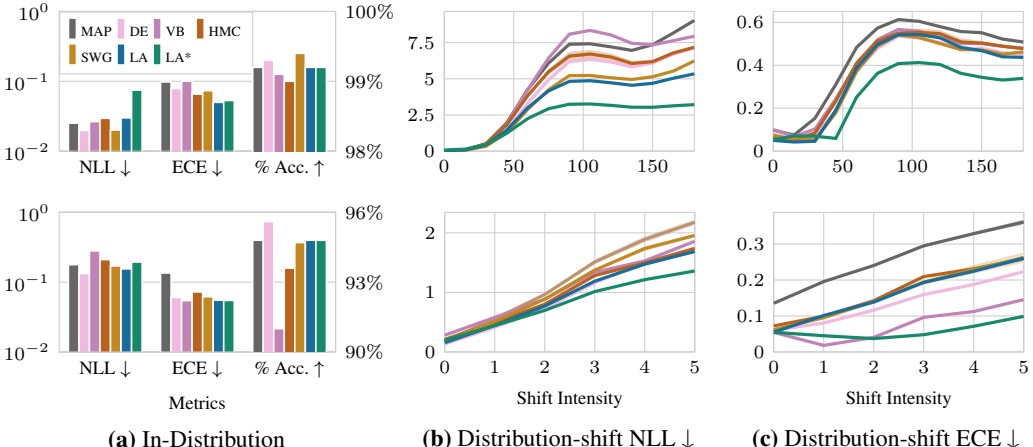

(a) In-Distribution     (b) Distribution-shift NLL ↓     (c) Distribution-shift ECE ↓

**Figure 4:** Assessing model calibration **(a)** on in-distribution data and **(b,c)** under distribution shift, for the MNIST (top row) and CIFAR-10 (bottom row) datasets. For **(b,c)**, we use the Rotated-MNIST (top) and Corrupted-CIFAR-10 (bottom) benchmarks [63, 64]. In **(a)**, we report accuracy and, to measure calibration, negative log-likelihood (NLL) and expected calibration error (ECE)—all evaluated on the standard test sets. In **(b)** and **(c)**, we plot shift intensities against NLL and ECE, respectively. For Rotated-MNIST (top), shift intensities denote degrees of rotation of the images, while for Corrupted-CIFAR-10 (bottom), they denote the amount of image distortion (see [63, 64] for details). **(a)** On in-distribution data, LA is the best-calibrated method in terms of ECE, while also retaining the accuracy of MAP (unlike VB and CSGHMC). **(b,c)** On corrupted data, all Bayesian methods improve upon MAP significantly. Even though *post-hoc*, all LAs achieve competitive results, even to DE. In particular, LA* achieves the best results, at the expense of slightly worse in-distribution calibration—this trade-off between in- and out-of-distribution performance has been observed previously [65].

predictive provides good performance in-distribution, but a probit or MC predictive improves OOD performance.

**Overall recommendation.** Following the experimental evidence, the default in `laplace` is a *post-hoc* KFAC last-layer LA with a GGN approximation to the Hessian. This default is applicable to all architectures that have a fully-connected last layer and can be easily applied to pre-trained networks. For problems where trained networks are unavailable or hyperparameters are unknown, the online KFAC LA with a GGN or empirical Fisher provides a good baseline with minimal effort.

### 4.2 Predictive Uncertainty Quantification

We consider two flavors of LAs: the default flavor of `laplace` (**LA**) and the most robust one in terms of distribution shift found in Section 4.1 (**LA***—last-layer, with a full empirical Fisher Hessian approximation, and the probit approximation). We compare them with the MAP network (**MAP**) and various popular and strong Bayesian baselines: Deep Ensemble [**DE**, 14], mean-field variational Bayes [**VB**, 11, 12] with the flipout estimator [66], cyclical stochastic-gradient Hamiltonian Monte Carlo [**CSGHMC / HMC**, 67], and SWAG [**SWG**, 15]. For each baseline, we use the hyperparameters recommended in the original paper—see Appendix A for details. First, Fig. 4 shows that **LA** and **LA*** are, respectively, competitive with and superior to the baselines in trading-off between in-distribution calibration and dataset-shift robustness. Second, Table 1 shows that **LA** and **LA*** achieve better results on out-of-distribution (OOD) detection than even **VB**, **CSGHMC**, and **SWG**.

The LA shines even more when we consider its (time *and* memory) cost relative to the other, more complex baselines. In Fig. 5 we show the wall-clock times of each method relative to **MAP**'s for training and prediction. As expected, **DE**, **VB**, and **CSGHMC** are slow to train and in making predictions: they are between two to five times more expensive than MAP. Meanwhile, despite being *post-hoc*, **SWG** is almost twice as expensive as **MAP** during training due to the need for sampling and updating its batch normalization statistics. Moreover, with 30 samples, as recommended by its authors [15], it is very expensive at prediction time—more than ten times more expensive than **MAP**.

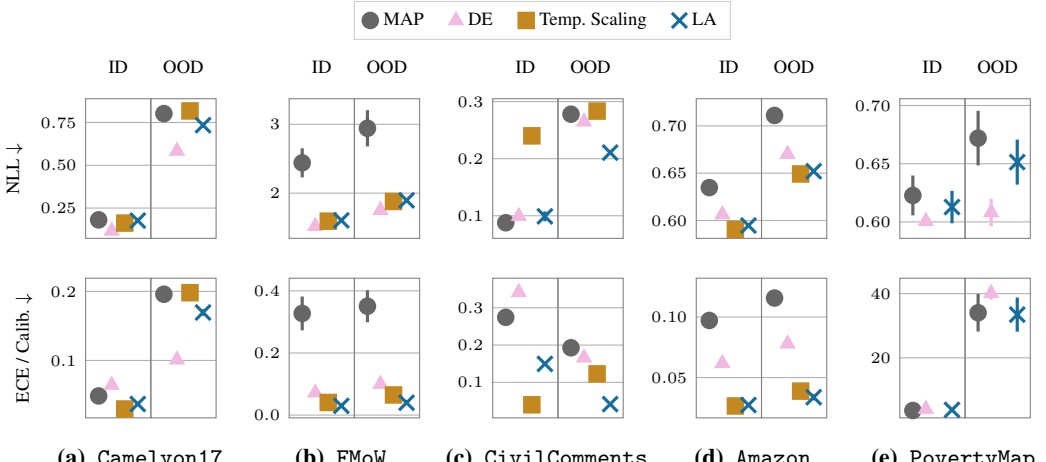

**Figure 6:** Assessing real-world distribution shift robustness on five datasets from the WILDS benchmark [68], covering different data modalities, model architectures, and output types. Camelyon17: Tissue slide image tumor classification across hospitals (DenseNet-121 [69]). FMoW: Satellite image land use classification across regions/years (DenseNet-121). CivilCommments: Online comment toxicity classification across demographics (DistilBERT [70]). Amazon: Product review sentiment classification across users (DistilBERT). PovertyMap: Satellite image asset wealth regression across countries (ResNet-18 [35]). We plot means $\pm$ standard errors of the NLL (top) and ECE (for classification) or regression calibration error [71] (bottom). The in-distribution (left panels) and OOD (right panels) dataset splits correspond to different domains (e.g. hospitals for Camelyon17). LA is much better calibrated than MAP, and competitive with temp. scaling and DE, especially on the OOD splits.

Meanwhile, **LA** (and **LA\***) is the cheapest of all methods considered: it only incurs a negligible overhead on top of the costs of **MAP**. This is similar for the memory consumption (see Table 5 in Appendix C.5). This shows that the LA is significantly more memory- and compute-efficient than all the other methods, adding minimal overhead over MAP inference and prediction. This makes the LA particularly attractive for practitioners, especially in low-resource environments. Together with Fig. 4 and Table 1, this justifies our default flavor in `laplace`, and importantly, shows that Bayesian deep learning does not have to be expensive.

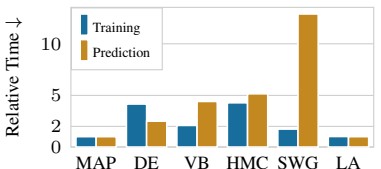

**Figure 5:** Wall-clock time costs relative to MAP. LA introduces negligible overhead over MAP, while all other baselines are significantly more expensive.

### 4.3 Realistic Distribution Shift

So far, our experiments focused on comparably simple benchmarks, allowing us to comprehensively assess different LA variants and compare to more involved Bayesian methods such as VB, MCMC, and SWAG. In more realistic settings, however, where we want to improve the uncertainty of complex and costly-to-train models, such as transformers [61], these methods would likely be difficult to get to work well and expensive to run. However, one might often have access to a pre-trained model, allowing for the cheap use of *post-hoc* methods such as the LA. To demonstrate this, we show how `laplace` can improve the distribution shift robustness of complex pre-trained models in large-scale settings. To this end, we use WILDS [68], a recently proposed benchmark of realistic distribution shifts encompassing a variety of real-world datasets across different data modalities and application domains. While the WILDS models employ complex (e.g. convolutional or transformer) architectures as feature extractors, they all feed into a linear output layer, allowing us to conveniently and cheaply apply the last-layer LA. As baselines, we consider: 1) the pre-trained MAP models [68], 2) *post-hoc* temperature scaling of the MAP models (for classification tasks) [1], and 3) deep ensembles [14].[5] More details on the experimental setup are provided in Appendix C.3. Fig. 6 shows the results on five

---

[5] We simply construct deep ensembles from the various pre-trained models provided by Koh et al. [68].

different `WILDS` datasets (see caption for details). Overall, Laplace is significantly better calibrated than MAP, and competitive with temperature scaling and ensembles, especially on the OOD splits.

## 4.4 Further Applications

Beyond predictive uncertainty quantification, the LA is useful in wide range of applications such as Bayesian optimization [37], bandits [72], active learning [34, 73], and continual learning [24]. The `laplace` library conveniently facilitates these applications. As an example, we demonstrate the performance of the LA on the standard continual learning benchmark with the Permuted-MNIST dataset, consisting of ten tasks each containing pixel-permuted MNIST images [74]. Figure 7 shows how the all-layer diagonal and Kronecker-factored LAs can overcome *catastrophic forgetting*. In this experiment, we update the LAs after each task as suggested by Ritter et al. [24] and improve upon their result by tuning the prior precision through marginal likelihood optimization during

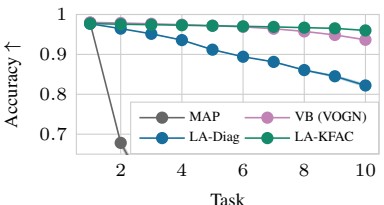

**Figure 7:** Continual learning results on Permuted-MNIST. MAP fails catastrophically as more tasks are added. The Bayesian approaches substantially outperform MAP, with LA-KFAC performing the best, closely followed by VOGN.

training, following Immer et al. [22] (details in Appendix C.4). Using this scheme, the performance after 10 tasks is at around 96% accuracy, outperforming other Bayesian approaches for continual learning [7, 75, 76]. Concretely, we show that the KFAC LA, while much simpler when applied via `laplace`, can achieve better performance to a recent VB baseline [VOGN, 13]. Our library thus provides an easy and quick way of constructing a strong baseline for this application.

## 5 Related Work

The LA is fundamentally a local approximation that covers a single mode of the posterior; similarly, other Gaussian approximations such as mean-field variational inference [11–13] or SWAG [15] also only capture local information. SWAG uses the first and second empirical moment of SGD iterates to form a diagonal plus low-rank Gaussian approximation but requires storing many NN copies and applying a (costly) heuristic related to batch normalization at test time. In contrast, the LA directly uses curvature information of the loss around the MAP and can be applied *post-hoc* to pre-trained NNs.

In contrast to local Gaussian approximations, (stochastic-gradient) MCMC methods [77, 78, 67, 79, 80, etc.] and deep ensembles [14] can explore several modes. Nevertheless, prior works—also validated in our experiments in Section 4—indicate that using a single mode might not be as limiting in practice as one might think. Wilson and Izmailov [81] conjecture that this is due to the complex, nonlinear connection between the parameter space and the function (output) space of NNs. Moreover, while unbiased compared to its simpler alternatives, MCMC methods are notoriously expensive in practice and, thus, often require further approximations such as distillation [82, 83]. Finally, note that both the LA as well as SWAG can be extended to ensembles of modes in a *post-hoc* manner [84, 81].

## 6 Conclusion

In this paper, we argued that the Laplace approximation is a simple yet competitive and versatile method for Bayesian deep learning that deserves wider adoption. To this end, we reviewed many recent advances to and variants of the Laplace approximation, including versions with minimal cost overhead that can be applied *post-hoc* to pre-trained off-the-shelf models. In a comprehensive evaluation we demonstrated that the Laplace approximation is on par with other approaches that approximate the intractable network posterior, but at typically much lower computational cost. A particularly simple variant that only treats some weights probabilistically can even be used in the context of pre-trained transformer models to improve predictive uncertainty. As an efficient implementation is not straightforward, we introduced `laplace`, a modular and extensible software library for PyTorch offering user-friendly access to all major flavors of the Laplace approximation. In this way, Laplace approximations provide drop-in Bayesian functionality for most types of deep neural networks.

## Acknowledgments and Disclosure of Funding

We thank Kazuki Osawa for providing early access to his automatic second-order differentiation (ASDL) library for PyTorch and Alex Botev for feedback on the manuscript. We also thank the anonymous reviewers for their helpful suggestions for our paper.

E.D. acknowledges funding from the EPSRC and Qualcomm. A.I. gratefully acknowledges funding by the Max Planck ETH Center for Learning Systems (CLS). R.E., A.K. and P.H. gratefully acknowledge financial support by the European Research Council through ERC StG Action 757275 / PANAMA; the DFG Cluster of Excellence "Machine Learning - New Perspectives for Science", EXC 2064/1, project number 390727645; the German Federal Ministry of Education and Research (BMBF) through the Tübingen AI Center (FKZ: 01IS18039A); and funds from the Ministry of Science, Research and Arts of the State of Baden-Württemberg. A.K. is grateful to the International Max Planck Research School for Intelligent Systems (IMPRS-IS) for support.

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
