# Appendix A    Derivation

## A.1    The Derivation of the Laplace Approximation

Let $p(\theta \mid \mathcal{D})$ be an intractable posterior, written as

$$p(\theta \mid \mathcal{D}) := \frac{1}{\int p(\mathcal{D} \mid \theta)p(\theta)\, d\theta} p(\mathcal{D} \mid \theta)p(\theta) =: \frac{1}{Z}h(\theta) \tag{1}$$

Our goal is to approximate this distribution with a Gaussian arising from the Laplace approximation. The key observation is that we can rewrite the normalizing constant $Z$ as the integral $\int \exp(\log h(\theta))\, d\theta$. Let $\theta_{\mathrm{MAP}} := \arg\max_\theta \log p(\theta \mid \mathcal{D}) = \arg\max_\theta \log h(\theta)$ be a (local) maximum of the posterior—the so-called ***maximum a posteriori (MAP)*** estimate. Taylor-expanding $\log h$ around $\theta_{\mathrm{MAP}}$ up to the second order yields

$$\log h(\theta) \approx h(\theta_{\mathrm{MAP}}) - \frac{1}{2}(\theta - \theta_{\mathrm{MAP}})^\top \Lambda\, (\theta - \theta_{\mathrm{MAP}}), \tag{2}$$

where $\Lambda := -\nabla^2 \log h(\theta)|_{\theta_{\mathrm{MAP}}}$ is the negative Hessian matrix of the log-joint in (1), evaluated at $\theta_{\mathrm{MAP}}$. Similar to its original formulation, here we again obtain a (multivariate) Gaussian integral, the analytic solution of which is readily available:

$$\begin{aligned} Z &\approx \exp(\log h(\theta_{\mathrm{MAP}})) \int \exp\left(-\frac{1}{2}(\theta - \theta_{\mathrm{MAP}})^\top \Lambda\, (\theta - \theta_{\mathrm{MAP}})\right)\, d\theta \\ &= h(\theta_{\mathrm{MAP}})\frac{(2\pi)^{\frac{d}{2}}}{(\det \Lambda)^{\frac{1}{2}}}. \end{aligned} \tag{3}$$

Plugging the approximations (2) and (3) back into the expression of $p(\theta \mid \mathcal{D})$, we obtain

$$p(\theta \mid \mathcal{D}) = \frac{1}{Z}h(\theta) \approx \frac{(\det \Lambda)^{\frac{1}{2}}}{(2\pi)^{\frac{d}{2}}} \exp\left(-\frac{1}{2}(\theta - \theta_{\mathrm{MAP}})^\top \Lambda\, (\theta - \theta_{\mathrm{MAP}})\right), \tag{4}$$

which we can immediately identify as the Gaussian density $\mathcal{N}(\theta \mid \theta_{\mathrm{MAP}}, \Sigma)$ with mean $\theta_{\mathrm{MAP}}$ and covariance matrix $\Sigma := \Lambda^{-1}$.

# Appendix B    Details on the Four Components

## ① Inference over Subsets of Weights

### B.1.1    Subnetwork

Storing the full $D \times D$ covariance matrix $\Sigma$ of the weight posterior in Eq. (4) is computationally intractable for a modern neural networks. One approach to reduce this computational burden is to perform inference over only a small *subset* of the model parameters $\theta$ [27]. This is motivated by recent findings that neural nets can be heavily pruned without sacrificing test accuracy [85], and that in the neighborhood of a local optimum, there are many directions that leave the predictions unchanged [46].

This *subnetwork inference* approach uses the following approximation to the posterior in Eq. (4):

$$p(\theta \mid \mathcal{D}) \approx p(\theta_S \mid \mathcal{D}) \prod_r \delta(\theta_r - \widehat{\theta}_r) = q_S(\theta), \tag{5}$$

where $\delta(x - a)$ denotes the Dirac delta function centered at $a$. The approximation $q_S(\theta)$ in Eq. (5) simply decomposes the full neural network posterior $p(\theta \mid \mathcal{D})$ into a Laplace posterior $p(\theta_S \mid \mathcal{D})$ over the subnetwork $\theta_S \in \mathbb{R}^S$, and fixed, deterministic values $\widehat{\theta}_r$ to the $D - S$ remaining weights $\theta_r$. In practice, the remaining weights $\theta_r$ are simply set to their MAP estimates, i.e. $\widehat{\theta}_r = \theta_r^{\mathrm{MAP}}$, requiring no additional computation. Importantly, note that the subnetwork size $S$ is in practice a hyperparameter that can be controlled by the user. Typically, $S$ will be set such that the subnetwork is much smaller than the full network, i.e. $S \ll D$. In particular, $S$ can be set such that it is tractable to compute and store the full $S \times S$ covariance matrix over the subnetwork. This allows us to capture rich

dependencies across the weights within the subnetwork. However, in principle one could also employ one of the (less expressive) factorizations of the Hessian/Fisher described in Section B.1.2.

Daxberger et al. [27] propose to choose the subnetwork such that the subnetwork posterior $q_S(\theta)$ in Eq. (5) is as close as possible (w.r.t. some discrepancy measure) to the full posterior $p(\theta \mid \mathcal{D})$ in Eq. (4). As the subnetwork posterior is degenerate due to the involved Dirac delta functions, common discrepancy measures such as the KL divergence are not well defined. Therefore, Daxberger et al. [27] propose to use the squared 2-Wasserstein distance, which in this case takes the following form:

$$W_2(p(\theta \mid \mathcal{D}), q_S(\theta))^2 = \text{Tr}\left(\Sigma + \Sigma_S - 2\left(\Sigma_S^{1/2}\, \Sigma\, \Sigma_S^{1/2}\right)^{1/2}\right), \tag{6}$$

where the (degenerate) subnetwork covariance matrix $\Sigma_S$ is equal to the full covariance matrix $\Sigma$ but with zeros at the positions corresponding to the weights $\theta_r$ (i.e. those *not* part of the subnetwork).

Unfortunately, finding the subset of weights $\theta_S \in \mathbb{R}^S$ of size $S$ that minimizes Eq. (6) is combinatorially hard, as the contribution of each weight depends on every other weight. Daxberger et al. [27] therefore assume that the weights are independent, resulting in the following simplified objective:

$$W_2(p(\theta \mid \mathcal{D}), q_S(\theta))^2 \approx \sum_{d=1}^{D} \sigma_d^2 (1 - m_d), \tag{7}$$

where $\sigma_d^2 = \Sigma_{dd}$ is the marginal variance of the $d^{\text{th}}$ weight, and $m_d = 1$ if $\theta_d \in \theta_S$ (with slight abuse of notation) or 0 otherwise is a binary mask indicating which weights are part of the subnetwork (see Daxberger et al. [27] for details). The objective in Eq. (7) is trivially minimized by choosing a subnetwork containing the $S$ weights with the highest $\sigma_d^2$ values (i.e. with largest marginal variances).

In practice, even computing the marginal variances (i.e. the diagonal of $\Sigma$) is intractable, as it requires storing and inverting the Hessian/Fisher $\Lambda$. To approximate the marginal variances, one could use a diagonal Laplace approximation [43, 2] that assumes $\text{diag}(\Sigma) \approx \text{diag}(\Lambda)^{-1}$. Alternatively, one could use diagonal SWAG [15]. For more details on subnetwork inference, refer to Daxberger et al. [27].

### B.1.2 Last-Layer

The last-layer Laplace [37, 28] is a special variant of the subnetwork Laplace where $\theta_S$ in (5) is assumed to equal the last-layer weight matrix $W^{(L)}$ of the network. That is, we let $f_\theta : \mathbb{R}^M \to \mathbb{R}^C$ is an $L$-layer NN, and assume that the first $L-1$ layers of $f_\theta$ is a feature map. Given MAP-trained parameters $\theta_{\text{MAP}}$, we define a Laplace-approximated posterior over $W^{(L)}$

$$p(W^{(L)} \mid \mathcal{D}) \approx \mathcal{N}(W^{(L)} \mid W_{\text{MAP}}^{(L)}, \Sigma^{(L)}), \tag{8}$$

and we leave the rest of the parameters with their MAP-estimated values. Since this matrix is small relative to the entire network, the last-layer Laplace can be implemented efficiently.

### ② Hessian Factorization

For brevity, given a datum $(x, y)$, we denote $s(x, y)$ to be the gradient of the log-likelihood at $\theta_{\text{MAP}}$, i.e.

$$s(x, y) := \nabla_\theta p(y \mid f_\theta(x))|_{\theta_{\text{MAP}}}.$$

Using this notation, we can write the Fisher compactly by

$$F := \sum_{n=1}^{N} \mathbb{E}_{p(y \mid f_\theta(x_n))}\left(s(x_n, y)s(x_n, y)^\intercal\right), \tag{9}$$

We shall refer to this matrix as the *full Fisher*. Recall that $F$ is as large as the exact Hessian of the network, so its computation is often infeasible. Thus, here, we review several factorization schemes that makes the computation (and storage) of the Fisher efficient, starting from the simplest.

**Diagonal** Although MacKay recommended to not use the diagonal factorization of the Hessian [86], a recent work has indicated this factorization is usable for sufficiently deep NNs [87]. In this factorization, we simply assume that the negative-log-posterior's Hessian $\Lambda$ is simply a diagonal matrix with diagonal elements equal the diagonal of the Fisher, i.e. $\Lambda \approx -\text{diag}(F)^\top I - \lambda I$. Since we can write $\text{diag}(F) = \sum_{n=1}^{N} \mathbb{E}_{p(y \mid f_{\theta_{\text{MAP}}}(x_n))}(s(x_n, y) \odot s(x_n, y))$,[6] this factorization is efficient: Not only does it require only a vector of length $D$ to represent $F$ but also it incurs only a $O(D)$ cost when inverting $\Lambda$—down from $O(D^3)$.

---

[6]The operator $\odot$ denotes the Hadamard product.

**KFAC** The KFAC factorization can be seen as a midpoint between the two extremes: diagonal factorization, which might be too restrictive, and the full Fisher, which is computationally infeasible. The key idea is to model the correlation between weights in the same layer but assume that any pair of weights from two different layers are independent—this is a more sophisticated assumption compared to the diagonal factorization since there, it is assumed that *all* weights are independent of each other. For any layer $l = 1, \ldots, L$, denoting $N_l$ as the number of hidden units at the $l$-th layer, let $W^{(l)} \in \mathbb{R}^{N_l \times N_{l-1}}$ be the weight matrix of the $l$-th layer of the network, $a^{(l)}$ the $l$-th hidden vector, and $g^{(l)} \in \mathbb{R}^{N_l}$ the log-likelihood gradient w.r.t. $a^{(l)}$. For each $l = 1, \ldots, L$, we can then write the outer product inside expectation in (8) as $s(x_i, y)s(x_i, y)^\top = a^{(l-1)}a^{(l)\top} \otimes g^{(l)}g^{(l)\top}$. Furthermore, assuming that $a^{(l-1)}$ is independent of $g^{(l)}$, we obtain the approximation of the $l$-th diagonal block of $F$, which we denote by $F^{(l)}$:

$$F^{(l)} \approx \mathbb{E}\left(a^{(l-1)}a^{(l-1)\top}\right) \otimes \mathbb{E}\left(g^{(l)}g^{(l)\top}\right) =: A^{(l-1)} \otimes G^{(l)}, \tag{10}$$

where we represent both the sum and the expectation in (9) as $\mathbb{E}$ for brevity.

From the previous expression we can see that the space complexity for storing $F^{(l)}$ is reduced to $O(N_l^2 + N_{l-1}^2)$, down from $O(N_l^2 N_{l-1}^2)$. Considering all $L$ layers of the network, we obtain the layer-wise Kronecker factors $\{A^{(l)}\}_{l=0}^{L-1}$ and $\{G^{(l)}\}_{l=1}^{L}$ of the log-likelihood's Hessian. This corresponds to the block-diagonal approximation of the full Hessian.

One can then readily use these Kronecker factors in a Laplace approximation. For each layer $l$, we obtain the $l$-th diagonal block of $\Lambda$—denoted $\Lambda^{(l)}$—by

$$\Lambda^{(l)} \approx \left(A^{(l-1)} + \sqrt{\lambda}I\right) \otimes \left(G^{(l)} + \sqrt{\lambda}I\right)$$
$$=: V^{(l)} \otimes U^{(l)}.$$

Note that we take the square root of the prior precision to avoid "double-counting" the effect of the prior. Nonetheless, this can still be a crude approximation [19, 26]. This particular Laplace approximation has been studied by Ritter et al. [23, 24] and can be seen as approximating the posterior of each $W^{(l)}$ with the matrix-variate Gaussian distribution [88]: $p(W^{(l)} | \mathcal{D}) \approx \mathcal{MN}(W^{(l)} | W_{\text{MAP}}^{(l)}, U^{(l)-1}, V^{(l)-1})$. Hence, sampling can be done easily in a layer-wise manner:

$$W^{(l)} \sim p\left(W^{(l)} | \mathcal{D}\right) \iff W^{(l)} = W_{\text{MAP}}^{(l)} + U^{(l)-\frac{1}{2}} E V^{(l)-\frac{1}{2}}$$

where

$$E \sim \mathcal{MN}(0, I_{N_l}, I_{N_{l-1}}),$$

where we have denoted by $I_b$ the identity $b \times b$ matrix, for $b \in \mathbb{N}$. Note that the above matrix inversions and square-root are in general much cheaper than those involving the entire $\Lambda$. Sampling $E$ is not a problem either since $\mathcal{MN}(0, I_{N_l}, I_{N_{l-1}})$ is equivalent to the standard $(N_l N_{l-1})$-variate Normal distribution. As an alternative, Immer et al. [26] suggest to incorporate the prior exactly using an eigendecomposition of the individual Kronecker factors, which can improve performance.

**Low-rank block-diagonal** We can improve KFAC's efficiency by considering its low-rank factorization [29]. The key idea is to eigendecompose the Kronecker factors in (10) and keep only the eigenvectors corresponding to the first $k$ largest eigenvalues. This can be done employing the eigenvalue-corrected KFAC [44]. That is, for each layer $l = 1, \ldots, L$:

$$F^{(l)} \approx \left(U_A^{(l-1)} S_A^{(l-1)} U_A^{(l-1)\top}\right) \otimes \left(U_G^{(l)} S_G^l U_G^{(l)\top}\right)$$
$$= \left(U_A^{(l-1)} \otimes U_G^{(l)}\right) \left(S_A^{(l-1)} \otimes S_G^{(l)}\right) \left(U_A^{(l-1)} \otimes U_G^{(l)}\right)^\top.$$

Under this decomposition, one can the easily obtain the optimal rank-$k$ approximation of $F^{(l)}$, denoted by $F_k^{(l)}$, by selecting the top-$k$ eigenvalues. However, the diagonal of this rank-$k$ matrix can deviate too far from the exact diagonal elements of $F^{(l)}$. Hence, one can make the diagonal of this low rank matrix exact replacing $\text{diag}(F_k^l)$ with $\text{diag}(F^{(l)})$, and obtain the following rank-$k$-plus-diagonal approximation of $F^{(l)}$:

$$F^{(l)} \approx F_k^{(l)} + \text{diag}(F^{(l)}) - \text{diag}(F_k^{(l)}).$$

This factorization can be seen as a combination of the previous two approximations: For each diagonal block of $F$, we use the exact diagonal elements of $F$ and approximate the off-diagonal elements with a rank-$k$ matrix arising from KFAC. Both the space and computational complexities are lower than those of KFAC since here we work exclusively with truncated and diagonal matrices.

**Low-rank**  Instead of only approximating each block by a low-rank structure, the entire Hessian or GGN can also be approximated by a low-rank structure [47, 46]. Eigendecomposition of $F$ is a convenient way to obtain a low-rank approximation. The eigendecomposition of $F$ is given by $QLQ^\top$ where the columns of $Q \in \mathbb{R}^{D \times D}$ are eigenvectors of $F$ and $L = \mathrm{diag}(l)$ is a $D$-dimensional diagonal matrix of eigenvalues. Assuming the eigenvalues in $l$ are arranged in a descending order, the optimal $k$-rank approximation in Frobenius or spectral norm is given by truncation [89]: let $\widehat{Q} \in \mathbb{R}^{D \times k}$ be the matrix of the first $k$ eigenvectors corresponding to the largest $k$ eigenvalues $\widehat{l} \in \mathbb{R}^k$. That is, we truncate all eigenvectors and eigenvalues after the $k$ largest eigenvalues. The low-rank approximation is then given by

$$F \approx \widehat{Q}\, \mathrm{diag}(\widehat{l})\, \widehat{Q}^\top.$$

The rank $k$ can be chosen based on the eigenvalues so as to retain as much information of the Hessian (approximation) as possible. Further, sampling and computation of the log-determinant can be carried out efficiently.

**Functional**  When considering network linearization for the predictive distribution, we can directly infer the Gaussian distribution on the outputs, of which there are typically few, instead of inferring a distribution on the parameters, of which there are many [25, 26].

## ③ Hyperparameter Tuning

In this section we focus on tuning the prior variance/precision hyperparameter for simplicity. The same principle can be used for other hyperparameters of the Laplace approximation such that observation noise in the case of regression.

***Post-Hoc***  Here, we assume that the steps of the Laplace approximation—MAP training and forming the Gaussian approximation—as two independent steps. As such, we are free to choose different prior variance $\gamma^2$ in the latter part, irrespective to the weight decay hyperparameter used in the former. Here, we review several ways to optimize $\gamma^2$ *post-hoc*. Ritter et al. [23] proposes to tune $\gamma^2$ by maximizing the posterior-predictive over a validation set $\mathcal{D}_{\mathrm{val}} := (x_n, y_n)_{n=1}^{N_{\mathrm{val}}}$. That is we solve the following one-parameter optimization problem:

$$\gamma_*^2 = \arg\max_{\gamma^2} \sum_{n=1}^{N_{\mathrm{val}}} \log p(y_n \mid x_n, \mathcal{D}). \tag{11}$$

However, Kristiadi et al. [28] found that the previous objective tends to make the Laplace approximation overconfident to outliers. Hence, they proposed to add an auxiliary term that depends on an OOD dataset $\mathcal{D}_{\mathrm{out}} := (x_n^{(\mathrm{out})})_{n=1}^{N_{\mathrm{out}}}$ to (11), as follows

$$\gamma_*^2 = \arg\max_{\gamma^2} \sum_{n=1}^{N_{\mathrm{val}}} \log p(y_n \mid x_n, \mathcal{D}) + \lambda \sum_{n=1}^{N_{\mathrm{out}}} H\left[ p(y_n \mid x_n^{(\mathrm{out})}, \mathcal{D}) \right], \tag{12}$$

where $H$ is the entropy functional and $\lambda \in (0, 1]$ is a trade-off hyperparameter. Intuitively, we choose $\gamma^2$ that balances the calibration on the true dataset and the low-confidence on outliers. Moreover, other losses could be constructed to tune the prior precision for optimal performance w.r.t. some desired quantity. Finally, inspired by Immer et al. [22] (further details below in *Online*) one can also maximize the Laplace-approximated marginal likelihood (3) to obtain $\gamma_*^2$, which eliminates the need for the validation data.

***Online***  Contrary to the *post-hoc* tuning above, here we perform a Laplace approximation and tune the prior variance simultaneously as we perform a MAP training [22]. The key is to form a Laplace-approximated posterior every $B$ epochs of a gradient descent, and use this posterior to approximate the marginal likelihood, cf. (3). By maximizing this marginal likelihood, we can find the best hyperparameters. Thus, once the MAP training has finished, we automatically obtain a prior variance that is already suitable for the Laplace approximation. Note that, this way, only a single MAP training needs to be done. This is in contrast to the classic, offline evidence framework [34] where

**Algorithm 1** Online Laplace (adapted from Immer et al. [22, Algorithm 1])

**Input:**

NN $f_\theta$; training set $\mathcal{D}$; learning rate $\alpha_0$ and number of epochs $T_0$ for MAP estimation; learning rate $\alpha_1$ and number of epochs $T_1$ for hyperparameter tuning; marginal likelihood maximization frequency $F$.

1:  Initialize $\theta_0$
2:  **for** $t = 1, \ldots, T_0$ **do**
3:      $g_t \leftarrow \nabla_\theta \mathcal{L}(\mathcal{D}; \theta)|_{\theta_{t-1}}$
4:      $\theta_t \leftarrow \theta_{t-1} - \alpha_0 \, g_t$
5:      **if** $t \mod F = 0$ **then**
6:          $p(\theta \,|\, \mathcal{D}) \approx \mathcal{N}(\theta \,|\, \theta_t, (\nabla^2 \mathcal{L}(\mathcal{D}; \theta)|_{\theta_t})^{-1})$        ▷ Perform a Laplace approximation
7:          **for** $\tilde{t} = 1, \ldots, T_1$ **do**        ▷ Hyperparameter optimization
8:              $h_{\tilde{t}} \leftarrow \nabla_{\gamma^2} \log p(\mathcal{D} \,|\, \gamma^2)|_{\gamma^2_{\tilde{t}-1}}$        ▷ The marginal likelihood follows from (3)
9:              $\gamma^2_{\tilde{t}} \leftarrow \gamma^2_{\tilde{t}-1} + \alpha_1 \, h_{\tilde{t}}$
10:          **end for**
11:      **end if**
12:  **end for**
13:  **return** $\theta_{T_0}$; $\nabla^2 \mathcal{L}(\mathcal{D}; \theta)|_{\theta_{T_0}}$

the marginal likelihood maximization is performed only when the MAP estimation is done, and these steps need to be iteratively done until convergence. As a final note, similar to the *post-hoc* marginal likelihood above, this *online Laplace* does not require a validation set and has an additional benefit of improving the network's generalization performance [22]. We refer the reader to Algorithm 1 for an overview.

## ④ Approximate Predictive Distribution

Here, we denote $x_* \in \mathbb{R}^N$ to be a test point, and $f_*$ be the network output at this point. We will review different way to approximate the predictive distribution $p(y \,|\, x_*, \mathcal{D})$ given a Gaussian approximate posterior, starting from the most general.

### B.4.3   General

**Monte Carlo Integration**    The simplest but general and unbiased approximation is the Monte Carlo (MC) integration, which can be performed by sampling an approximate posterior $q(\theta \,|\, \mathcal{D})$ repeatedly:

$$p(y \,|\, x_*, \mathcal{D}) \approx \frac{1}{S} \sum_{s=1}^{S} p(y \,|\, f_{\theta_s}(x_*)), \qquad \text{where } \theta_s \sim q(\theta \,|\, \mathcal{D}).$$

While the error of this approximation decays like $1/\sqrt{S}$ and thus requires many samples to be accurate, for practical BNNs, it is standard to use 10 or 20 samples of $q(\theta \,|\, \mathcal{D})$ [23, 28, 12, etc.]. Note that this approximation can be used regardless the form of the likelihood $p(y \,|\, f_\theta(x))$, in particular it can be used to directly obtain the predictive distribution in both the regression and classification alike.

### B.4.4   Distribution of Network Outputs

Here, we are concerned in approximating the marginal distribution of $f(x_*)$, where $\theta$ has been integrated out.

**Linearization**    In this approximation, we linearize the network to obtain

$$f_\theta(x_*) \approx f_{\theta_{\text{MAP}}}(x_*) + J_*^\top (\theta - \theta_{\text{MAP}}),$$

where $J_* := \nabla_\theta f_\theta(x_*)|_{\theta_{\text{MAP}}} \in \mathbb{R}^{d \times c}$ is the Jacobian matrix of the network output. This way, under a Gaussian approximate posterior $q(\theta \,|\, \mathcal{D})$, the marginal distribution over the network output $f_* :=$

$f(x_*)$ is again a Gaussian, given by[7]

$$p(f_* \mid f_\theta(x_*), x_*, \mathcal{D}) = \int \delta(f_* - f_\theta(x_*)) \, q(\theta \mid \mathcal{D}) \, d\theta$$

$$\approx \mathcal{N}(f_* \mid f_{\theta_{\mathrm{MAP}}}(x_*), J_*^\top \Sigma J_*)$$

This approximation has been extensively used for small networks [34], but it has since gone out of favor in deep learning due to its cost—the Jacobian $J_*$ needs to be computed *per input point*. Nevertheless, this approximation is still useful in theoretical works due to its analytical nature [28, 49, 84] . Moreover, in problems where it can be efficiently use in practice, it offers a better approximation than MC-integral [26, 48]. Due to the linearization in the network parameters, it is further possible to obtain a functional prior in the form of a Gaussian process [25, 26]. This allows to perform function-space inference as opposed to weight-space inference which is amenable to different Hessian approximations than those pointed out above in Section B.1.2, and is, for example, useful for continual learning [76].

### B.4.5 Regression

Assume that we already have a Gaussian approximation to $p(f_* \mid x_*, \mathcal{D}) \approx \mathcal{N}(f_* \mid \mu_*, \Sigma_*)$ via the linearization above. In regression, we still need to incorporate the observation noise $\beta$ encoded in the (usually) Gaussian likelihood $\mathcal{N}(y_* \mid f_*, \beta I)$[8] to make prediction. This can be easily done in an exact manner:

$$p(y_* \mid x_*) = \int_{\mathbb{R}^C} \mathcal{N}(y_* \mid f_*, \beta I) \, \mathcal{N}(f_* \mid \mu_*, \Sigma_*) \, df_*$$

$$= \mathcal{N}(y_* \mid \mu_*, \Sigma_* + \beta I),$$

since the integral above is just a convolution of two Gaussian r.v.s.

### B.4.6 Classification and Generalized Regression

Since unlike the regression case, the classification likelihood $p(y_* \mid f_*)$ is non-Gaussian, we cannot analytically obtain $p(y_* \mid x_*)$ given a Gaussian approximation $p(f_* \mid x_*, \mathcal{D}) \approx \mathcal{N}(f_* \mid \mu_*, \Sigma_*)$. So, in this case we are interested in approximating the intractable integral

$$p(y_* \mid x_*) = \int p(y_* \mid f_*) \, \mathcal{N}(f_* \mid \mu_*, \Sigma_*) \, df_*,$$

where $p(y_* \mid f_*)$ is constructed via an inverse-link function. Here we will review the usual case of classification, i.e. when $p(y_* \mid f_*) = \sigma(f_*)$ where $\sigma$ is the logistic-sigmoid function, or $p(y_* \mid f_*) = \mathrm{softmax}(f_*)$ .

**Delta Method**  The crux of the delta method [91–93] is a Taylor-expansion of the softmax function around $\mu_*$ up to the second order. Then, since $p(f_* \mid x_*, \mathcal{D})$ is assumed to be Gaussian, the integral $\mathbb{E}_{p(f_* \mid x_*, \mathcal{D})}(\mathrm{softmax}(f_*))$ can be computed easily, resulting in an analytic expression $\mathrm{softmax}(\mu_*) + 1/2 \, \mathrm{tr}(B\Sigma_*)$, where $B$ is the Hessian matrix of the softmax at $\mu_*$.

**Probit Approximations**  The essence of the (binary) probit approximation [31, 34] is to approximate $\sigma$ with the probit function $\Phi$—the standard Normal c.d.f.—which makes the integral solvable analytically. Using this approximation, one can then obtain the closed-form approximation

$$p(y_* \mid x_*) \approx \int_{\mathbb{R}} \Phi(f_*) \, \mathcal{N}(f_* \mid \mu_*, \sigma_*^2) \, df_*$$

$$= \sigma \left( \frac{\mu_*}{\sqrt{1 + \frac{\pi}{8} \sigma_*^2}} \right) .$$

It has a generalization to multi-class classification, due to Gibbs [52], i.e. for approximating

$$p(y_* \mid x_*) = \int_{\mathbb{R}^C} \mathrm{softmax}(f_*) \, \mathcal{N}(f_* \mid \mu_*, \Sigma_*) \, df_*. \tag{13}$$

---

[7]See Bishop [90, Sec. 4.5.2].
[8]We assume a multivariate output $y_* \in \mathbb{R}^C$ for full generality.

|          | test log likelihood | test accuracy | OOD-AUROC | prediction time (s) |
|----------|---------------------|---------------|-----------|---------------------|
| DIAG     | -0.302±0.005        | 0.894±0.002   | 0.832±0.011 | 29.5±0.2          |
| KFAC     | -0.282±0.004        | 0.899±0.002   | 0.836±0.004 | 30.6±0.1          |
| FULL     | -0.285±0.004        | 0.898±0.002   | 0.876±0.003 | 62.8±1.1          |

**Table 2:** Qualitative comparison of different Hessian approximations. The KFAC Hessian approxima-
tion performs similar to FULL Gauss-Newton but is almost as fast as DIAG. We use online marginal
likelihood method [22] to train a small convolutional network on FMNIST and measure performance
at test time. We repeat for three seeds to estimate the standard error. The OOD-AUROC is averaged
over EMNIST, MNIST, and KMNIST. The prediction time is taken as the average over all in and
out-of-distribution data sets. We use the MC predictive with 100 samples.

In this case, we approximate the resulting probability vector of length $C$ with a vector which $i$-th
component is given by $\exp(\tau_i)/\sum_{j=1}^{C} \exp(\tau_j)$, where $\tau_j = \mu_{*j}/\sqrt{1 + \pi/8\, \Sigma_{*jj}}$ for each $j = 1, \ldots, C$. This approximation ignores the correlation between logits since it only depends on the
diagonal of $\Sigma_*$. Nevertheless, it yields good results even in deep learning [65], and are invaluable
tools for theoretical work [84].

**Laplace Bridge**    The main idea of the Laplace bridge is to perform a Laplace approximation to
the Dirichlet distribution by first writing it as a distribution over $\mathbb{R}^C$ with the help of the softmax
function [53, 54]. This way, Laplace approximation can be reasonably applied to approximate the
Dirichlet, which can be thought as mapping the Dirichlet $\mathrm{Dir}(\alpha_*)$ to a Gaussian $\mathcal{N}(\mu_*, \Sigma_*)$. The
pseudo-inverse of this map, mapping $(\mu_*, \Sigma_*)$ to $\alpha_*$ where for each $i = 1, \ldots, C$, the $i$-th component
$\alpha$ is given by the simple closed-form expression

$$\alpha_i = \frac{1}{\Sigma_{ii}} \left( 1 - \frac{2}{C} + \frac{\exp(\mu_i)}{C^2} \sum_{j=1}^{C} \exp(-\mu_j) \right),$$

is the *Laplace bridge*. Just like the probit approximation, the Laplace bridge ignores the correlation
between logits. But, unlike all the previous approximations, it yields a *full distribution* over the
solutions of the softmax-Gaussian integral (13). So, the Laplace bridge is a richer yet comparably
simple approximation to the integral and is useful for many applications in deep BNNs [55].

## Appendix C   Further Experiments Details and Results

### C.1   Laplace Comparison

Here, we present more detailed results of our comparison of the different variations of the Laplace
approximation. We show in-distribution accuracy for CIFAR-10 using a model trained with and
without data augmentation, and AUROC values averaged over the out-of-distribution datasets SVHN,
LSUN, and CIFAR-100. In the first row of Figure 8, we highlight the different Hessian structures with
different colors; in the second row, we use color to highlight the different link approximations in the
predictive distribution. We considered most combinations of the different choices for the components
discussed in Section 2, but exclude some combinations which we have found to not work well at all,
e.g. online Laplace when performing a Laplace approximation over the weights of only the last layer.
In Table 2, we compare the predictive performance and runtime when using differently structured
Hessian approximations. We find that the Kronecker-factored Hessian approximations provides a
good trade-off between runtime and performance.

### C.2   Predictive Uncertainty Quantification

#### C.2.1   Training Details

We use LeNet [94] and WideResNet-16-4 [WRN, 95] architectures for the MNIST and CIFAR-10
experiments, respectively. We adopt the commonly-used training procedure and hyperparameter
values.

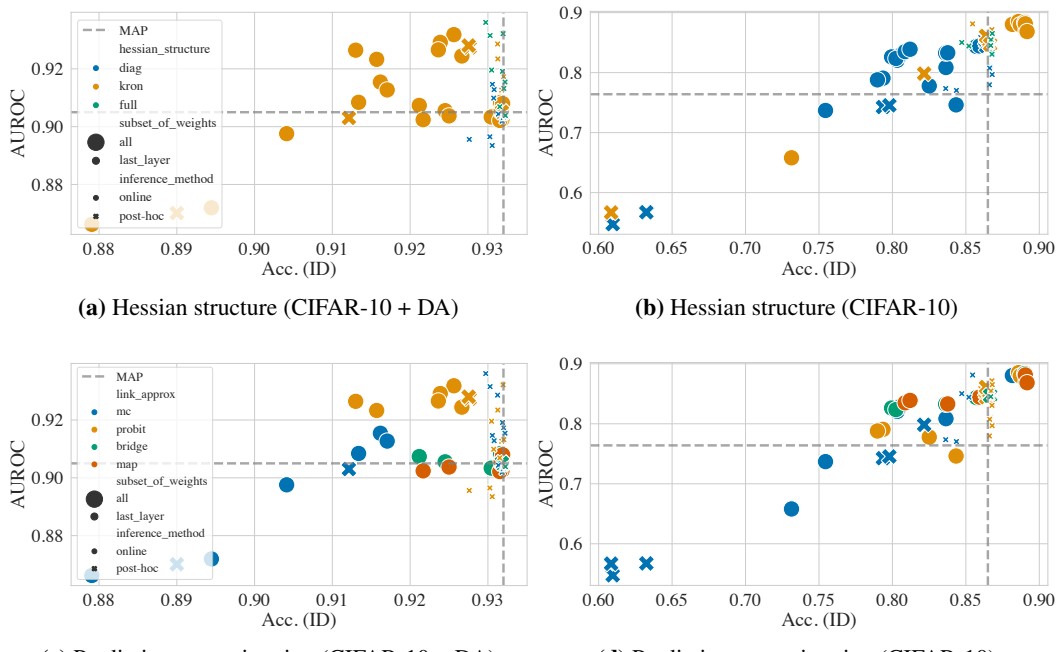

**(a)** Hessian structure (CIFAR-10 + DA)

**(b)** Hessian structure (CIFAR-10)

**(c)** Predictive approximation (CIFAR-10 + DA)

**(d)** Predictive approximation (CIFAR-10)

**Figure 8:** Comparison of variations of the LA on the CIFAR-10 OOD experiment with ((a) and (c)) and without ((b) and (d)) data augmentation (DA).

**MAP**   We use Adam and Nesterov-SGD to train LeNet and WRN, respectively. The initial learning rate is $0.1$ and annealed via the cosine decay method [96] over $100$ epochs. The weight decay is set to $5 \times 10^{-4}$. Unless stated otherwise, all methods below use these training parameters.

**DE**   We train five MAP network (see above) independently to form the ensemble.

**VB**   We use the Bayesian-Torch library [97] to train the network. Tha variational posterior is chosen to be the diagonal Gaussian [11, 12] and the flipout estimator [66] is employed. The prior precision is set to $5 \times 10^{-4}$ to match the MAP-trained network, while the KL-term downscaling factor is set to $0.1$, following [13].

**CSGHMC**   We use the publicly available code provided by the original authors [67].[9] We use their default (i.e. recommended) hyperparameters.

**SWAG**   For the SWAG baseline, we follow Maddox et al. [15] and run stochastic gradient descent with a constant learning rate on the pre-trained models to collect one model snapshot per epoch, for a total of 40 snapshots. At test time, we then make predictions by using 30 Monte Carlo samples from the posterior distribution; we correct the batch normalization statistics of each sample as described in Maddox et al. [15]. To tune the constant learning rate, we used the same approach as in Eschenhagen et al. [84], combining a grid search with a threshold on the mean confidence. For MNIST, we defined the grid to be the set { 1e-1, 5e-2, 1e-2, 5e-3, 1e-3 }, yielding an optimal value of 1e-2. For CIFAR-10, searching over the same grid suggested that the optimal value lies between 5e-3 and 1e-3; another, finer-grained grid search over the set { 5e-3, 4e-3, 3e-3, 2e-3, 1e-3 } then revealed the best value to be 2e-3.

**Other baselines**   Our choice of baselines is based on the most common and best performing methods of recent Bayesian DL papers. Despite its popularity, **Monte Carlo (MC) dropout** [6] has been shown to underperform compared to more recent methods (see e.g. Ovadia et al. [64]). A recent VI method called **Variational Online Gauss-Newton (VOGN)** [13] also seems to underperform. For example, Fig. 5 of Osawa et al. [13] shows that on OOD detection with CIFAR-10 vs. SVHN, MC-dropout and

[9]https://github.com/ruqizhang/csgmcmc

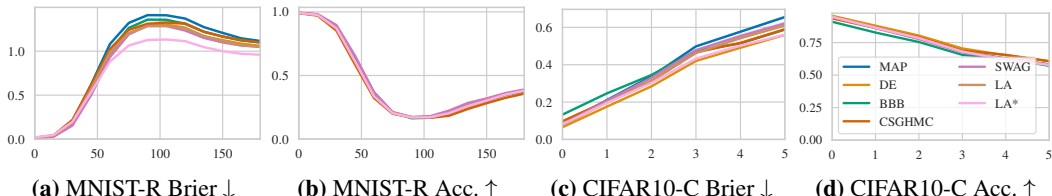

| (a) MNIST-R Brier ↓ | (b) MNIST-R Acc. ↑ | (c) CIFAR10-C Brier ↓ | (d) CIFAR10-C Acc. ↑ |

**Figure 9:** Dataset shift on the Rotated-MNIST (top) and Corrupted-CIFAR-10 datasets (bottom).

**Table 3:** MNIST OOD detection results.

| Methods | Confidence ↓ | | | AUROC ↑ | | |
|---|---|---|---|---|---|---|
| | EMNIST | FMNIST | KMNIST | EMNIST | FMNIST | KMNIST |
| MAP | 83.6±0.3 | 64.2±0.5 | 77.3±0.3 | 93.5±0.3 | 98.9±0.0 | 97.0±0.1 |
| DE | 75.8±0.2 | 55.4±0.4 | 65.9±0.3 | 95.1±0.0 | 99.2±0.0 | 98.3±0.0 |
| BBB | 79.1±0.4 | 67.5±1.6 | 73.1±0.4 | 92.3±0.2 | 98.2±0.2 | 97.0±0.2 |
| CSGHMC | 76.2±1.6 | 63.6±1.9 | 67.9±1.5 | 93.4±0.2 | 97.7±0.2 | 97.1±0.1 |
| SWAG | 64.9±0.3 | 84.0±0.2 | 78.5±0.3 | 98.9±0.0 | 93.6±0.3 | 97.1±0.1 |
| LA | 74.8±0.4 | 58.8±0.5 | 69.0±0.4 | 93.4±0.3 | 98.5±0.1 | 96.6±0.1 |
| LA* | 62.0±0.5 | 49.6±0.6 | 56.7±0.5 | 94.3±0.2 | 98.3±0.1 | 96.6±0.2 |

VOGN only achieve AUROC↑ values of 81.9 and 80.0, respectively, while last-layer-LA obtains a substantially better value of 91.9 (they use ResNet-18, which is comparable to our model).

### C.2.2 Detailed Results

We show the Brier score and accuracy as a function of shift intensity in Fig. 9. Moreover, we provide the detailed (i.e. non-averaged) OOD detection results in Tables 3 and 4.

### C.2.3 Additional Details on Wall-clock Time Comparison

Concerning the wall-clock time comparison in Fig. 5, we would like to clarify that for LA, we consider the default configuration of `laplace`. As the default LA variant uses the closed-form probit approximation to the predictive distribution and therefore neither requires Monte Carlo (MC) sampling nor multiple forward passes, the wall-clock time for making predictions is essentially the same as for MAP. This is contrast to the baseline methods, which are significantly more expensive at prediction time due to the need for MC sampling (VB, SWAG) or forward passes through multiple model snapshots (DE, CSGHMC).

Importantly, note that is an advantage exclusive to our implementation of LA (i.e. with a GGN/Fisher Hessian approximation or with the last-layer LA) that it can be used without sampling (i.e. using the probit or Laplace bridge predictive approximations). This kind of approximation is incompatible with the other baselines (i.e. DE, CSGHMC, SWAG, and VB) since these methods just yield samples/distributions over weights while our LA variants implicitly yield a Gaussian distribution over logits due to the linearization of the NN induced by the use of the GGN/Fisher (see Immer et al. [26] for details) or the use of only the last layer. While one could still apply linearization to other methods, this would not be theoretically justified, in contrast to GGN-/last-layer-LA.

Finally, the reason we benchmark our deterministic, probit-based version is that we found it to consistently perform on par or better than MC sampling. If we predict with the LA using MC samples on the logits, the runtime is only around 20% slower than the deterministic probit approximation, which is still significantly faster than all other methods.

In summary, we believe that the ability to obtain calibrated predictions with a single forward-pass is a critical and distinctive advantage of the LA over almost all other Bayesian deep learning and ensemble methods.

**Table 4:** CIFAR-10 OOD detection results.

| Methods | Confidence ↓ | | | AUROC ↑ | | |
|---|---|---|---|---|---|---|
| | **SVHN** | **LSUN** | **CIFAR-100** | **SVHN** | **LSUN** | **CIFAR-100** |
| MAP | 77.5±2.9 | 71.3±0.6 | 79.3±0.1 | 91.8±1.2 | 94.5±0.2 | 90.1±0.1 |
| DE | 62.8±0.7 | 62.6±0.4 | 70.8±0.0 | 95.4±0.2 | 95.3±0.1 | 91.4±0.1 |
| BBB | 60.2±0.7 | 53.8±1.1 | 63.8±0.2 | 88.5±0.4 | 91.9±0.4 | 84.9±0.1 |
| CSGHMC | 69.8±0.8 | 65.2±0.8 | 73.1±0.1 | 91.2±0.3 | 92.6±0.3 | 87.9±0.1 |
| SWAG | 69.3±4.0 | 62.2±2.3 | 73.0±0.4 | 91.6±1.3 | 94.0±0.7 | 88.2±0.5 |
| LA | 70.6±3.2 | 63.8±0.5 | 72.6±0.1 | 92.0±1.2 | 94.6±0.2 | 90.1±0.1 |
| LA* | 58.0±3.1 | 50.0±0.5 | 59.0±0.1 | 91.9±1.3 | 95.0±0.2 | 90.2±0.1 |

## C.3 `WILDS` Experiments

For this set of experiments, we use `WILDS` [68], a recently proposed benchmark of realistic distribution shifts encompassing a variety of real-world datasets across different data modalities and application domains. In particular, we consider the following `WILDS` datasets:

- `Camelyon17`: Tumor classification (binary) of histopathological tissue images across different hospitals (ID vs. OOD) using a DenseNet-121 model (10 seeds).

- `FMoW`: Building / land use classification (62 classes) of satellite images across different times and regions (ID vs. OOD) using a DenseNet-121 model (3 seeds).

- `CivilCommments`: Toxicity classification (binary) of online text comments across different demographic identities (ID vs. OOD) using a DistilBERT-base-uncased model (5 seeds).

- `Amazon`: Sentiment classification (5 classes) of product reviews across different reviewers (ID vs. OOD) using a DistilBERT-base-uncased model (3 seeds).

- `PovertyMap`: Asset wealth index regression (real-valued) across different countries and rural/urban areas (ID vs. OOD) using a ResNet-18 model (5 seeds).

Please refer to the original paper for more details on this benchmark and the above-mentioned datasets. All reported results in Fig. 6 and Fig. 10 show the mean and standard error across as many seeds as there are provided with the original paper (see the list of datasets above for the exact numbers).

For the last-layer Laplace method, we use either a KFAC or full covariance matrix (depending on the size of the last layer; in particular, we use a KFAC covariance for `FMoW` and full covariances for all other datasets) and the linearized Monte Carlo predictive distribution with 10,000 samples.

For the deep ensemble, we simply the aggregate the pre-trained models provided by the original paper[10] This yields ensembles of 5 neural network models, which is a commly-used ensemble size [64]. Since these models were trained in different ways (e.g. using different domain generalization methods, see [68] for details), their combinations can be viewed as *hyperparameter ensembles* [98].

Note that the temperature scaling baseline is only applicable for classification tasks, and therefore we do not report it for the `PovertyMap` regression dataset.

We tune the temperature parameter for temperature scaling, the prior precision parameter for Laplace, and the noise standard deviation parameter for regression (i.e. for the `PovertyMap` dataset) by minimizing the negative log-likelihood on the in-distribution validation sets provided with `WILDS`.

Finally, Fig. 10 shows an extended version of the results reported in Fig. 6, which additionally reports the following metrics: accuracy (for classification) or mean squared error (for regression), confidence (only for classification), mean calibration error (only for classification), and Brier score (only for classification). The overall conclusion here is the same as for Fig. 6, namely that Laplace is significantly better calibrated than MAP, and competitive with temperature scaling and ensembles, especially on the OOD splits. Note that the differences in accuracies of the ensemble stem from the different training procedures of the ensemble members (which sometimes achieve higher and sometimes lower accuracy), as mentioned above.

---

[10]See `https://worksheets.codalab.org/worksheets/0x52cea64d1d3f4fa89de326b4e31aa50a` for the complete list of models.

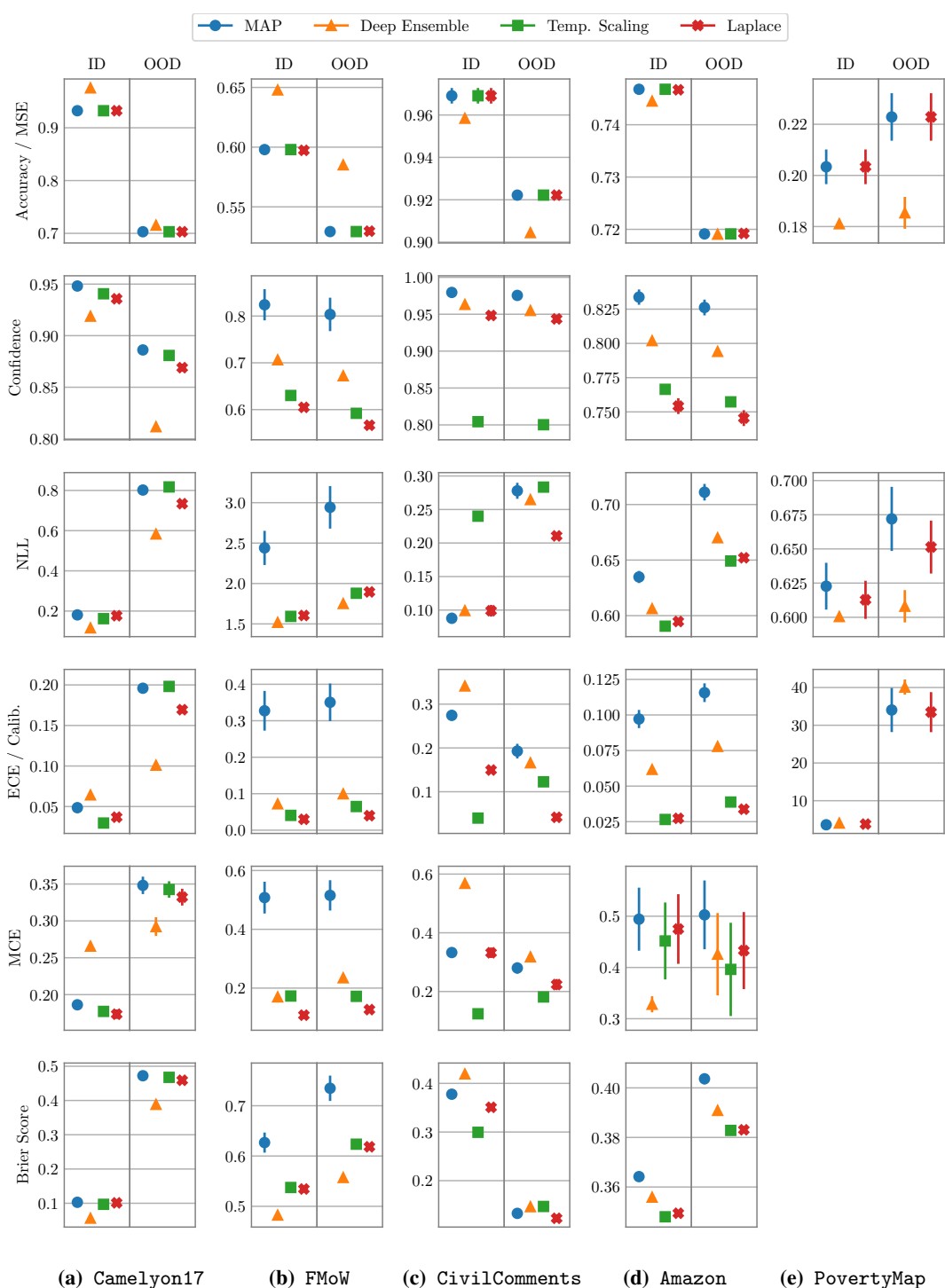

**Figure 10:** Assessing real-world distribution shift robustness on five datasets from the WILDS benchmark [68], covering different data modalities, model architectures, and output types; see text for details. We report means ± standard errors of several metrics (from top to bottom): accuracy (for classification) or mean squared error (for regression), confidence (only for classification), negative log-likelihood, ECE (for classification) or regression calibration error [71], mean calibration error (only for classification), and Brier score (only for classification). The in-distribution (left panels) and OOD (right panels) dataset splits correspond to different domains (e.g. hospitals for Camelyon17).

## C.4 Further Details on the Continual Learning Experiment

We benchmark Laplace approximations in the Bayesian continual learning setting on the *permuted MNIST* benchmark which consists of 10 consecutive tasks where each task is a permutation of the pixels of the MNIST images. Following common practice [24, 7, 13], we use a 2-hidden layer MLP with 100 hidden units each and $28 \times 28 = 784$ input dimensions and 10 output dimensions for the MNIST classes. We adopt the implementation of the continual learning task and the model by Pan et al. [76].[11] In the following, we will briefly outline the Bayesian approach to continual learning [7] and explain how a diagonal and KFAC Laplace approximation can be employed in this setting. Further, we describe how this can be combined with the evidence framework to update the prior online alleviating the need for a validation set, which is unlikely to be available in real continual learning scenarios.

### C.4.1 Bayesian Approach to Continual Learning

The Bayesian approach to continual learning can be simply described as iteratively updating the posterior after each task. We are given $T$ data sets $\mathcal{D} := \{\mathcal{D}_t\}_{t=1}^T$ and have a neural network with parameters $\theta$. In line with the standard supervised learning setting outlined in Section 2, we have a prior on parameters $p(\theta) = \mathcal{N}(\theta; 0, \gamma^2 I)$ and a likelihood $p(\mathcal{D} \mid \theta)$ realized by a neural network. The posterior on the parameters after all tasks is then

$$p(\theta \mid \mathcal{D}) \propto p(\mathcal{D}_T \mid \theta) \times \ldots \times p(\mathcal{D}_2 \mid \theta) \times \underbrace{p(\mathcal{D}_1 \mid \theta) \times p(\theta)}_{\substack{\propto p(\theta \mid \mathcal{D}_1) \\ \propto p(\theta \mid \mathcal{D}_1, \mathcal{D}_2)}}. \tag{14}$$

This factorization gives rise to a recursion to update the posterior after $t-1$ data sets to the posterior after $t$ data sets:

$$p(\theta \mid \mathcal{D}_1, \ldots, \mathcal{D}_t) \propto p(\mathcal{D}_t \mid \theta) p(\theta \mid \mathcal{D}_1, \ldots, \mathcal{D}_{t-1}). \tag{15}$$

The normalizer for each update in Eq. (15) is given by the marginal likelihood $p(\mathcal{D}_t \mid \mathcal{D}_1, \ldots, \mathcal{D}_{t-1})$ and we will use it for optimizing the variance $\gamma^2$ of $p(\theta)$. Incorporating a new task is the same as Bayesian inference in the supervised case but with an updated prior, i.e., the prior is the previous posterior distribution on $\theta$. The Laplace approximation provides one way to approximately infer the posterior distributions after each task [99, 24, 76]. Alternatively, variational inference can be used [7, 13].

### C.4.2 The Laplace Approximation for Continual Learning

The Laplace approximation facilitates the recursive updates (Eq. (15)) that arise in continual learning. In this context, it was first suggested with a diagonal Hessian approximation by Kirkpatrick et al. [2, EWC] and Huszár [99] corrected their updates. Ritter et al. [24] greatly improved the performance by using a KFAC Hessian approximation instead of a diagonal. The Laplace approximation to the posterior after observing task $t$ is a Gaussian $\mathcal{N}(\theta_{\text{MAP}}^{(t)}, \Sigma^{(t)})$ We obtain $\theta_{\text{MAP}}$ by optimizing the unnormalized log posterior distribution on $\theta$ as annotated in Eq. (14) for every task, one after another. The Hessian of the same unnormalized log posterior also specifies the posterior covariance $\Sigma^{(t)}$:

$$\Sigma^{(t)} = \left( \underbrace{\nabla_\theta^2 \log p(\mathcal{D}_t \mid \theta)|_{\theta_{\text{MAP}}^{(t)}}}_{\text{log likelihood Hessian}} + \underbrace{\sum_{t'=1}^{t-1} \nabla_\theta^2 \log p(\mathcal{D}_{t'} \mid \theta)|_{\theta_{\text{MAP}}^{(t')}}}_{\text{previous log likelihood Hessians}} + \underbrace{\gamma^{-2} I}_{\text{log prior Hessian}} \right)^{-1}. \tag{16}$$

This summation over Hessians is typically intractable for neural networks with large parameter vectors $\theta$ and hence diagonal or KFAC approximations are used [2, 99, 24]. For the diagonal version, the addition of Hessians and log prior is exact. For the KFAC version, we follow the alternative suggestion by Ritter et al. [24] and add up Kronecker factors which is an approximation to the sum of Kronecker products. However, this approximation is what underlies KFAC even in the supervised learning case where we add up factors per data point over the entire data set. Lastly, we adapt $\gamma$ during training on each task $t$ by optimizing the marginal likelihood $p(\mathcal{D}_t \mid \mathcal{D}_1, \ldots \mathcal{D}_{t-1})$, i.e., by differentiating it with respect to $\gamma$. This can be done by computing the eigendecomposition of the summed Kronecker factors [22] and allows us to 1) adjust the regularization suitably per task and 2) avoid setting a hyperparameter thereby alleviating the need for validation data.

---

[11]The code is avilable at `https://github.com/team-approx-bayes/fromp`.

**Table 5:** The memory complexities of all methods in $\mathcal{O}$ notation. To get a better idea of what these complexities translate to in practice, we also report the actual memory footprints (in megabytes) of a Wide ResNet 16-4 (WRN) on CIFAR-10. Here, $M$ denotes the number of model parameters, $H$ denotes the number of neurons in the last layer, $K$ denotes the number of model outputs, $R$ denotes the number of SWAG snapshots, $S$ denotes the number of CSGHMC samples, and $N$ denotes the number of deep ensemble (DE) members. Mean-field variational inference (VB) has a complexity of $2M$ as it needs to store a variance vector of size $M$ in addition to the mean vector of size $M$. For the actual memory footprints, we assume $R = 40$ SWAG snapshots, $S = 12$ CSGHMC samples, and $N = 5$ ensemble members, which are the hyperparameters recommended in the original papers (and therefore also used in our experiments). It can be seen that the proposed default KFAC-last-layer approximation poses a small memory overhead of $\mathcal{O}(H^2 + K^2)$ on top of the MAP estimate.

| METHOD | MEM. COMPLEXITY | WRN ON CIFAR-10 |
|---|:---:|:---:|
| MAP | $M$ | 11MB |
| **LA** | $\mathbf{M + H^2 + K^2}$ | **12MB** |
| VB | $2M$ | 22MB |
| DE | $NM$ | 55MB |
| CSGHMC | $SM$ | 132MB |
| SWAG | $RM$ | 440MB |

## C.5 Comparison of Memory Complexity

Table 5 compares the theoretical memory complexity and actual memory footprint (of a Wide ResNet 16-4 on CIFAR-10) of the different methods.