# OpenReview forum: "Laplace Redux - Effortless Bayesian Deep Learning"
_NeurIPS.cc/2021/Conference — NeurIPS 2021 Poster_

### Official Review · Reviewer_R8DH · 2021-07-08

**Rating:** 7
**Confidence:** 3

**Summary:**

The paper considers the problems of approximate Bayesian inference in deep learning. The paper argues that the Laplace approximation does not receive the attention it deserves in the community and that the Laplace approximation is a good trade-off between simplicity, computational complexity, and predictive performance. The paper then provides a review of the relevant theory and the relevant decision for setting up Laplace approximations in deep learning. The main contribution of the paper is a software package for computing Laplace approximations based on pre-trained model. The paper concludes with a large empirical study comparing Laplace approximations to a range of other relevant methods.


**Main Review:**

The paper addresses an important problem - how to make Bayesian inference feasible for large-scale deep networks? I really enjoyed reading the paper. While the paper does not contain much novelty in terms of methodological development, I believe this paper and the software package could be of great interest to many researchers and practitioners in the field.

The first sections of the paper (introduction, background on the Laplace approx, description of the software) are in generally very well-structured, well-written, and easy to follow. However, I found the structure of the experimental section quite confusing and it took me a while to decipher the information because of this. For example, the authors refer to Figs. 3, 4 and 5 in section 4.1, but all the abbreviations used in the figures are not introduced before section 4.2. Besides, there are several experimental details that are unclear to me (see comments below). Therefore, it think the experimental section could benefit significantly from a restructuring.

The claims of the paper are substantiated to a reasonable degree based on theoretical and empirical evidence from the literature as well as by a fairly comprehensive empirical study based on several different datasets. However, it would have been even more interesting if the authors had chosen to include more adhoc methods like Monte Carlo Dropout, etc., and more advanced VI methods for completeness. But that being said, I think the empirical results are already quite extensive.



Other comments:

- Figure 3: What does the individual dots in Fig. 3 represent?

- Figure 4: It is still a bit unclear what exactly the axis in (b) and (c) are showing. Please elaborate on this. How is the shift intensity defined and what is on the y-axis?

- What is the definition of "Confidence" defined in Table 1?

- In several experiments, the authors compared results for models with fixed hyperparameters vs learned hyperparameters. However, it is very unclear to me which parameters the authors consider to be in the set of hyperparameters. For example, which quantities are considered as hyperparameters in Fig 3? And how are they initialized when learning them through the approximate marginal likelihood?

- The paper provides a set of general recommendations based on the specific experiments. It would be nice if the authors would include a discussion of how well those recommendations will generalize to other types of problems.

- In line 97 the authors argue that it can be sufficient to approximate a subset of weights using the Laplace approximation because it has been shown that some networks can be pruned without sacrificing test accuracy. However, I fail to see how this justifies only modeling uncertainty for a subset of the weights. When pruning networks, you don't simply ignore the uncertainty - you completely remove the weights. Could the authors please elaborate a bit on this?

**Post rebuttal**
Updated score from 6 to 7.

**Time Spent Reviewing:**

2

---

> ### Author Response · Authors · 2021-08-10
> **Response to Reviewer R8DH**
>
> We thank you very much for your detailed and helpful feedback! In the following, we address your main review and answer your questions. We hope these address your concerns and convince you to reconsider your score. If not, please do let us know so we can clarify things further. Please also note the general response we posted, which clarifies some important points regarding our contributions.
>
> ### Comments to main review
> **Experimental section structure confusing.** We take your point that parts of the experimental section are hard to parse because it is very dense. We plan to save some space on the background section to clarify the experimental section.
>
> **Missing baseline methods.** Our choice of baselines is based on the most common and best performing methods of recent Bayesian DL papers. For example, MC-dropout has often been shown to underperform compared to more recent methods (see e.g. [Ovadia et al. 2019](https://arxiv.org/abs/1906.02530)). Below we show a preliminary comparison between last-layer-LA, MC-dropout (MCD), and a more recent VI method ([VOGN](https://arxiv.org/abs/1906.02506)) on OOD detection with CIFAR-10 vs. SVHN (measured via AUROC, where higher is better). The results for MC-dropout and VOGN are taken from Figure 5 of [Osawa et al. 2019](https://arxiv.org/abs/1906.02506) (they use a comparable, though not identical, network architecture to our paper, i.e. ResNet-18). The table shows that LA is significantly better than MC-dropout and VOGN. We will add a more comprehensive comparison to the paper.
>
> | Method | AUROC |
> |--------|-------|
> | MCD    | 81.9  |
> | VOGN   | 80.0  |
> | **LA**     | **91.9**  |
>
> ### Answers to questions and comments
> **Figure 3.** Each dot in Fig. 3 represents one combination of design choices (i.e. one particular setting of 1) subset-of-weights, 2) covariance structure, 3) parameter tuning, and 4) predictive). For a more exhaustive annotation, see Appendix C.1 and Figure 8.
>
> **Figure 4.** For Corrupted-CIFAR-10 (bottom), the shift-intensities refer to the amount of distortion applied to the images; this is best illustrated visually: please see Figure 7 in [Hendrycks and Dietterich 2019](https://arxiv.org/abs/1903.12261) (who introduced this benchmark) and Figure S2 in [Ovadia et al. 2019](https://arxiv.org/abs/1906.02530) (who first used this benchmark in an uncertainty quantification context). For Rotated-MNIST (top), the shift-intensities refer to the degrees of rotation applied to the images; see e.g. Figure S1 (a) in [Ovadia et al. 2019](https://arxiv.org/abs/1906.02530) for a visual illustration. We will make this clearer and elaborate more on it in the appendix. The y-axes show negative log-likelihood (NLL) and expected calibration error (ECE) as indicated in captions (b) and (c) with corresponding arrows indicating that lower is better for both. The idea is that ideally a model is robust to distribution shift and therefore still has a good low negative test log likelihood (NLL; i.e. predicts correctly) and maintains a good calibration in terms of ECE (i.e. knows when it predicts incorrectly).
>
> **Confidence in Table 1.** It is defined as the maximum of the probability vector of the prediction (for example, if the model predicts [0.7, 0.2, 0.1] for three classes, then the confidence would be 0.7). See also [Hendrycks and Gimpel 2016](https://arxiv.org/abs/1610.02136).
>
> **Which hyperparameters are considered?** For hyperparameters, we consider the regularization strength (== prior precision) individually per layer of the neural network for the “online” method and a single scalar regularization strength for “post-hoc”. When learning through the marginal likelihood, we initialize the hyperparameters to $1$ (not normalized by data set) for simplicity but the algorithm is robust to this choice as long as the initial regularization is not too strong and drives weights immediately to $0$. We will add these details to the experimental details in the appendix.
>
> **Subnetwork Laplace vs. pruning.** We meant to intuitively explain the methodology by [Daxberger et al. 2021](http://proceedings.mlr.press/v139/daxberger21a.html) by this analogy. In fact, they do not prune weights but only consider a subset of weights to perform inference over. This can be seen as a generalization of last-layer LA, which fixes the subset of weights to be from the last layer. We will make this clearer in the paper.

---

> > ### Comment · Reviewer_R8DH · 2021-08-27
> > **Updating score to accept**
> >
> > Thank you for the message and the additional details. You carefully addressed most of my comments. If the additional details and clarification will be added to the paper, then I'll recommend acceptance of the paper.

---

> > > ### Author Response · Authors · 2021-08-27
> > > **Thank you very much!**
> > >
> > > Thank you very much for updating your score. We will include all additional details and clarifications in the final version of the paper.

---

### Official Review · Reviewer_oDXG · 2021-07-09

**Rating:** 7
**Confidence:** 4

**Summary:**

The authors bring to light the recent progress in approximate Bayesian inference via the Laplace approximation, and note that despite its simplicity, its adoption has not kept pace for modern deep learning methods. The work describes a consolidation of recent work into a user-friendly PyTorch-based software library called `laplace`. LA can be applied post-hoc to existing deep learning models. With this package in place, the authors provide practical recommendations for reasonable defaults when used in practice.

**Main Review:**

I think this paper is very well-timed given the modern evidence of Laplace approximations for Bayesian inference being effective. As the authors identify, despite its conceptual simplicity, the developments have not kept pace with modern times in terms of easy-to-use software packages. In that sense, this is much appreciated. I have specific comments and question below.

- In terms of writing, while the introduction is clear, but for a paper that aims to convince the reader about the simplicity of implementing scalable LA, Section 2 does not inspire confidence. I think this may only be a matter of organizing and contextualizing the approximations available better.

- To me, the most important part of the paper is the experiments section. I think the most important aspect missing in the study is the sensitivity of the approximations. How sensitive is the final performance to the Hessian approximations? Clearly, this has to be consequential?

- Something that would also be interesting to see with this broad-scoped analysis is whether we see the Occam's razor effect during the marginal likelihood optimization still happen even with the approximation?

- Another important point worth noting is that Multi-SWAG observed improvements using mutli-modal approximations of SWAG. Have the authors investigated this with Laplace approximation? Given that Laplace approximation is a finer approximation of the rather empirical curvature approximation with SWAG, do we really get gains there? Is it worth using LA to more finely capture geometry?

Overall, this is well-timed paper to remind the community of important progress in scalable LA. I do think, however, that there are major missed opportunities in terms of the qualitative analysis of the method beyond benchmarking (which is no doubt diverse and reasonably exhaustive), especially in terms of the sensitivity and other qualitative improvements that LA provides over Deep Ensembles or SWAG. I will vote for a weak accept, but I would be more than happy to be convinced otherwise since this is an important effort towards consolidation of scalable LA.

### Minor Comments

- For Figure 3, it may be helpful to the eye to have both graphs with the same scales to make comparison visually easier.
- It would be good have a detailed caption summarizing the message in Figure 4 (and if space is a constraint, perhaps move bulk of the explanation to the figure).
- Lines 31-32 should mention SWAG [1], which appears to satisfy good empirical performance at the cost of a single network. The authors do compare to SWAG later as well anyways.

[1]:: https://arxiv.org/abs/1902.02476

---

**UPDATE**: See comment below. Updating score from 6 to 7.

**Time Spent Reviewing:**

5

---

> ### Author Response · Authors · 2021-08-10
> **Response to Reviewer oDXG**
>
> We thank you a lot for your helpful and thorough feedback! Below are our answers to your questions and comments (in the same order). We are eager to clarify and discuss things further if this does not convince you enough to raise your score. Please also note the general response we posted, which clarifies some important points regarding our contributions.
>
> - **Section 2 (background) too complicated.** We could make this section more accessible by 1) making the high-level picture (Figure 2) clearer and 2) move detailed specifics to a self-contained section in the appendix. Would that be helpful?
>
> - **Sensitivity of the approximations.** We study the sensitivity to choice of Hessian approximation and other choices in Section 4.1 with further details in Appendix C.1. In short, we find that better approximations tend to improve performance. However, a full/dense approximation provides minimal benefit over KFAC and is in most cases intractable. This observation motivates KFAC variants as default choice. To demonstrate this, we conducted an experiment comparing the KFAC and diagonal variants to more expensive Hessian approximations (i.e. dense full-network Hessian) on a sufficiently small CNN on FashionMNIST -- on the architectures we considered in our paper this is unfortunately not tractable due to our focus on common deep learning architectures. The table below reports the test log-likelihoods (LL, higher is better), test accuracies (Acc, higher is better) and AUROC scores for OOD detection (averaged across all OOD datasets; higher is better) for one run with online marginal likelihood tuning and the full-network MC predictive. We also report the test inference time in seconds (averaged across the in-distribution and OOD datasets). These results show that the full/dense Hessian approximation is *significantly slower* than the diagonal and KFAC variants while not offering a tremendous performance benefit (only in terms of AUROC). We will add a more comprehensive evaluation on this to the paper -- thanks a lot for this suggestion!
>
> | Method | LL     | Acc   | AUROC | Time (s)|
> |--------|--------|-------|-------|---------|
> | Diag   | -0.293 | 0.899 | 0.842 |  30.67  |
> | KFAC   | -0.276 | 0.903 | 0.836 |  35.22  |
> | Full   | -0.279 | 0.902 | 0.878 | 137.85  |
>
>
> - **Occam’s razor effect with the approximate marginal likelihood.** Please refer to [Immer et al. 2021](https://arxiv.org/abs/2104.04975) who introduced the method that we implement. They found that an Occam-type effect can be observed even with diagonal Hessian approximations in the case of large neural networks.
>
> - **Multi-modal LA.** Multi-modal LA (multi-LA) is a very good idea. In fact, one can construct a multi-LA similar to multi-SWAG, as done in [75]. They find that this improves over a single LA and therefore significantly improves over deep ensembles and multi-SWAG in terms of performance and runtime. Please find below an extension of the OOD detection results in Table 1, showing the confidence (Conf., lower is better) and AUROC (higher is better) scores for multi-SWAG and multi-LA. The results show that multi-LA outperforms / is competitive with multi-SWAG (both outperform the deep ensemble, which is the strongest method in Table 1). However, importantly, multi-LA is *significantly cheaper* both in terms of runtime (cf. Figure 5 in our paper; multi-{LA,SWAG} naturally yield a linear increase in runtime over their unimodal counterparts, similar as with MAP vs DE) and memory (see the table with memory complexities we report in our response to reviewer T5ck; again, the multi-variants incur a factor of $N$ = number of ensemble members). Finally, note that implementing multi-LA is *very easy* with `laplace`.
>
> | Method     | Conf. (MNIST)  | Conf. (CIFAR10) | AUROC (MNIST)  | AUROC (CIFAR10) |
> |------------|----------------|-----------------|----------------|-----------------|
> | Multi-SWAG | $65.9 \pm 0.3$ | $59.6 \pm 0.9$  | $97.6 \pm 0.0$ | $94.1 \pm 0.3$  |
> | Multi-LA   | $56.1 \pm 0.4$ | $53.9 \pm 0.5$  | $97.9 \pm 0.0$ | $95.0 \pm 0.1$  |
>
> - **Minor comments.** We agree regarding the figures and will make the corresponding changes. We will also refer to SWAG earlier but want to emphasize that SWAG is *significantly more expensive* than a single network, especially for making predictions (see Figure 5 and also the table of memory complexities in our response to reviewer T5ck).

---

> > ### Comment · Reviewer_oDXG · 2021-08-11
> > **Updating score to recommend a clear accept**
> >
> > Thank you for addressing the comments in detail. In light of new evidence, I do not see a reason why I should hold back my score. I am positive that the authors will be able polish the writing (including Section 2 as suggested in their response above).

---

> > > ### Author Response · Authors · 2021-08-27
> > > **Thank you very much!**
> > >
> > > Thank you very much for updating your score. We will polish the writing (including Section 2 as suggested in our response) in the final version of the paper.

---

### Official Review · Reviewer_T5ck · 2021-07-15

**Rating:** 6
**Confidence:** 5

**Summary:**

The paper discusses Laplace approximations in the context of approximate Bayesian inference for deep neural networks.
The Authors propose a new library, laplace, that allow practitioners to easily extend loss-trained models (MAP) to Bayesian.
The new library allows also the user to choose different approximations for the Hessian, different strategies for the Bayesian extension (e.g. by considering all or just a subset of weights) and a variety of procedures to choose/learn the hyper-parameters.


**Limitations And Societal Impact:**

Limitations are only partially addressed.
In my opinion the is one important discussion that is missing and it regards the memory consumption, for CPU and GPU implementations.
All Bayesian methods are known to be extremely challenging for both timing and memory usage: the Authors do a good job from the timing perspective (see Figure 4) but they neglect to address this other point of view.
As a future practitioner, especially if I'm working with constrained resources, e.g. very small GPU or edge devices, I would be very interested in a proper comparison between approximations/performance versus memory usage.
This would be very helpful to really understand all the different trade-off I should consider for if I intend to use LA.


**Main Review:**


Methodologically the contributions of this work are very thin as the Laplace approximation is a well known and simple way to do approximate Bayesian inference.
Even the application of LA to deep neural networks is not new (see [21,25] in paper).
At the same time, I agree with the Authors when they argue on the poor adoption rate of LA, especially when compared against more popular variational and MCMC methods, and it's true that this might be due to missing reference implementations easily available.

For this reason, I have to positively evaluate the empirical benchmark, the analysis of different approximations and the source code (which is well documented and which promises to be very user-friendly).
Realistically, this paper will not be groundbreaking in terms of contributions but it provides good baselines for future research in BDL.

I was playing a bit with the code and the 1D regression example and overall it looks easy to tinker the different choices/parameters.
I have found a major limiting design choice of the library, though: unless I've missed it in the code/documentation, it seems that only Gaussian prior can be used. Indeed it looks like the Gaussian prior is hard-coded in implementation of the approximation (and of the marginal likelihood).
In practice, there is evidence that heavy-tail priors (like student-t) can deliver better performance than Gaussian [a,b], and the fact that there is no option for the user to choose the prior is a problem.
Same comment applies also for choice of the likelihood (only Gaussian and multi-class).

## Minor
One minor remark: it would be better to have a minimal version for the requirements. I had an older version of PyTorch installed and some functions were not available, causing the script to fail (the setuptools didn't update the packages due to the missing minimal version).

[a] Tran et al. All you need is a good functional prior.
[b] Fortuin et al. Bayesian Neural Network Priors Revisited


**Time Spent Reviewing:**

10

---

> ### Author Response · Authors · 2021-08-10
> **Response to Reviewer T5ck**
>
> Thanks a lot for your thorough and insightful feedback! We hope the following points address your concerns and convince you to raise your score. If not, please do let us know by replying to this post. We would be happy to clarify things further. Please also note the general response we posted, which clarifies some important points regarding our contributions.
>
> ### Likelihoods and priors
> Generally, we plan to extend the library regarding priors and likelihoods to accommodate for more use cases. In particular, it would be interesting to see how different priors affect the LA in line with the study by Fortuin et al. who only studied this question using SGLD.
>
> **Likelihoods.** For the initial version, we chose to implement only Categorical and Gaussian likelihoods as these cover most (if not all) current benchmarks, and Bayesian deep learning is at this point in time largely focused on classification and regression settings. It is possible to extend to other likelihoods in the future though. This would have to happen in conjunction with extending the backends which focus on the same likelihoods. Do you have any particular likelihood in mind that you think would be interesting to include?
>
> **Priors.** In fact, any prior can be used in the current version of the library because `laplace`  only requires passing the Hessian of the log-prior. For example, Student-t would be straightforward this way. Nevertheless, you are right that the marginal likelihood currently only supports Gaussian priors. That being said, we are already working on supporting more priors in a more user-friendly way. However, this was not of top priority since studies regarding the prior are still partly inconclusive and Gaussian priors are still the most common. For example, [Izmailov et al. 2021](https://arxiv.org/pdf/2104.14421.pdf) recently found with large-scale HMC that Gaussian priors are sufficient.
>
> ### Limitations
> We agree that a memory comparison is missing and will add it -- thanks a lot for this suggestion! In terms of complexities, the proposed default KFAC-last-layer approximation poses a small memory overhead of $O(H^2+K^2)$ on top of the MAP estimate, where $H$ is the number of neurons in the last layer and $K$ the number of outputs. Please find below a table of memory complexities for various methods, where $M$ denotes the number of model parameters, $R$ denotes the number of SWAG snapshots, $S$ denotes the number of CSGHMC samples, and $N$ denotes the number of deep ensemble (DE) members. Mean-field variational inference (VB) has a complexity of $2M$ as it needs to store a variance vector of size $M$ in addition to the mean vector of size $M$. To get a better idea of what these complexities translate to in practice, we also report the actual memory footprints of a Wide ResNet 16-4 (WRN) on CIFAR10, assuming $R=40$ SWAG snapshots, $S=12$ CSGHMC samples, and $N=5$ ensemble members, which are the hyperparameters recommended in the original papers (and therefore also used in our experiments).
>
> | Method | Mem. Complexity | WRN on CIFAR10  |
> |--------|-----------------|-----------------|
> | MAP    | $M$             | 11 MB           |
> | **LA**     | **$M + H^2 + K^2$** | **12 MB**           |
> | VB     | $2M$            | 22 MB           |
> | DE     | $NM$            | 55 MB           |
> | CSGHMC | $SM$            | 132 MB          |
> | SWAG   | $RM$            | 440 MB          |
>
> As shown in the table, **the LA is significantly more memory-efficient** than all the other methods and is therefore preferable in low-resource environments because it adds minimal overhead over MAP inference and prediction. The LA is the only approximation with this feature, making it **particularly attractive for practitioners**.

---

> > ### Comment · Reviewer_T5ck · 2021-08-30
> > **Keep my score**
> >
> > First of all, many thanks for the reply. I wanted to wait for the discussion with the other reviewers before making my final decision. The main concern for me remains the flexibility of the library, especially in the choices of likelihoods and priors. I believe that with future releases this can be easily fixed, but unfortunately at this time this is not enough for me to increase the score.

---

> > > ### Author Response · Authors · 2021-09-01
> > > **Thank you for your response!**
> > >
> > > Dear reviewer, thank you very much for your response and again for your insightful review, which will help us improve our paper and library!

---

### Official Review · Reviewer_vbUc · 2021-07-16

**Rating:** 6
**Confidence:** 3

**Summary:**

This paper describes a software library for scalable Laplace approximations in PyTorch, enabling more efficient implementations of all of its various forms and variants (from KFAC approximations to marginal likelihood optimization to different approximations at test time).

**Ethical Concerns:**

No.

**Limitations And Societal Impact:**

Yes.

**Main Review:**

tldr: I tend to weakly reject this paper because I'm really ambivalent about accepting what is essentially a research software package with experiments.

## Originality

*Are the tasks or methods new?* Not really, the software package is. The methods themselves have been around for a while.

*Is the work a novel combination of well-known techniques?* The software package is.

*Is it clear how this work differs from previous contributions?* The authors do a good job of distinguishing their implementation from others.

*Is related work adequately cited?* Mostly, ref [71] is mis-cited and uses HMC (although they compare to SGMCMC as well), while the GGMC method of [Garriga-alonso and Fortuin, AABI, '20](https://arxiv.org/abs/2102.01691) probably also deserves a citation due to its recent popularity.


## Quality

*Is the submission technically sound?* Yes, I think so. The software package seems pretty well put together.

*Are claims well supported (e.g., by theoretical analysis or experimental results)?*
- Fig 5 is a pretty unfair comparison, as, deep ensembles, HMC and SWAG are using sampling (either during train / test or both), while the Laplace approximations studied use deterministic predictions.
- The comparisons between (HMC, SWAG, and LA*) seem a lot more mixed for OOD detection from Table 1 than is depicted in the paper. It looks like LA* is mostly significantly less confident (some type of extra tuning?) than any method, but that this doesn't translate into better AUROC.

*Are the methods used appropriate?* Mostly so given this space, a few comments:
- I would have preferred to see comparisons to full Hessian / Fisher Laplace approximations on say MNIST since this should be doable (implement scalable Hessian vector or Fisher vector products and then use a low rank approximation). This shouldn't be too difficult in the authors' software framework and might be a convincing demonstration of the modularity a bit more.

From what I can see, you either use a low rank KFAC Fisher approximation or a last layer one. A dense approximation would be helpful to see if it provides improved benefits.

- Given the high amount of modularity in MC approximation, one pretty clear missing piece of comparison is that of probit approximation + SWAG (or Laplace bridge + SWAG but it Laplace bridge seems to be dominated by other methods so less necessary). After all, in Fig 5 the wall-clock time is devoted to the computational cost of the number of samples...
- The comparisons to CSGHMC in the appendix should absolutely be moved to the main text, and shouldn't be buried that deep.

*Is this a complete piece of work or work in progress?* Complete work.

*Are the authors careful and honest about evaluating both the strengths and weaknesses of their work?* Pretty much.

## Significance

*Are the results important?* Yes, if this gains steam, then it'll probably be very useful.

*Are others (researchers or practitioners) likely to use the ideas or build on them?* Yes, releasing a software package for scalable laplace approximations should definitely be useful.

*Does the submission address a difficult task in a better way than previous work?* Not really.

*Does it advance the state of the art in a demonstrable way?* Yes, I think so.

*Does it provide unique data, unique conclusions about existing data, or a unique theoretical or experimental approach?*

## Discussion & Questions

I'm very ambivalent about accepting this paper because almost the entire contribution is the software package. Included in the citations are references for research works that do, give or take maybe one experiment, most of the comparisons in this paper. My score is pretty much based on this ambivalence, with some concerns about comparisons addressed above.

Thinking some more about this, there's fairly few research software packages (essentially) that end up published particularly because it's so difficult to evaluate software packages. From my understanding, the few packages that have ended up published ([Autoconj](https://arxiv.org/abs/1811.11926), [neural-tangents](https://openreview.net/forum?id=SklD9yrFPS)) have included what is effectively state of the art engineering capabilities as part of their contributions.

Overall, I really like the message of the paper and think that it could be a boon to the community to see this type of demonstration published. First, I like the comparisons demonstrating that something as straightforward as Laplace approximations (and really even last layer ones) can easily outperform variational methods. Second, I like the demonstration that MC sampling is often outperformed by deterministic probit style approximations.

### Writing / Minor Comments

- I really dislike that I'm saying this, but please editorialize your captions some more. Include a bit of takeaway for each figure caption --- at least "what is the one sentence summary of this figure?"
- Example: Fig 7. Permuted MNIST: all Bayesian approaches outperform MAP estimation in continual learning, with LA-KFAC performing the best, closely followed by VOGN.
- The caption on Fig 3 is especially rough.


-------

## Review of Software Package

After discussion with the area chair, I decided to take a more detailed look at the software package, which seems to be of reasonably high quality. A quick review of the package itself is here:

I thank the authors for taking the time to develop the package and for putting a bunch of different Laplace approximations all in the same place in a mostly modular fashion. However, I think that it's still pretty oriented around research code, rather than practicability, but this might be a personal feeling. Ultimately, I'm still not sure that people would use it outside of their current implementations of Hessian vector products or their current implementations of KFAC, without a lot of advertisement, although as a researcher, I'd probably take a look at it.

- Can you please describe what the (presumably anonymized) backend `asdfghjkl` is?
- While the implementation is described as modular, from a peek through the package, it's not clear to me how one would implement full rank Hessians via say the torch functional interface (https://pytorch.org/docs/stable/generated/torch.autograd.functional.hvp.html), rather than backpack? Would this require setting up an entirely different backend in order to do so?
- Are there any restrictions on layers that the backpack backend has? I recall that originally backpack couldn't handle some types of convolutional layers (I believe).
- As a comment, I find that the package is rather focused on KFAC style approximations, rather than other types of approximations (for example, those based around low rank approximations of the Hessian / Fisher defined via matrix vector multiplications or even layer-wise versions of these). Thus, it seems tricky to see how to extend beyond these types of approximations, making the package more limited in scope than is promised.
- `torch.kron` is in pytorch now, might want to update to that.





**Time Spent Reviewing:**

4

---

> ### Author Response · Authors · 2021-08-10
> **Response to Reviewer vbUc**
>
> Thank you very much for your constructive and detailed feedback! We hope the following points help resolve your ambivalence regarding our work and convince you to reconsider your score. We address all your points below. For a detailed response regarding the relevance of our work to NeurIPS and our contributions beyond the library, please see our general response. If things remain unclear, please do let us know by replying to this post. We are happy to discuss things further.
>
> ## Originality
> Please see our general response for details on secondary contributions, including a new benchmark and new methods.
> Thanks for pointing out the problem with our SGMCMC reference and the missing reference to the work of Garriga-Alonso and Fortuin. We will fix that.
>
> ## Quality
> **Figure 5 [comparing runtimes] is an unfair comparison.** We respectfully disagree. It is an advantage exclusive to our implementation of LA (i.e. with a GGN/Fisher Hessian approximation or with the last-layer LA) that it can be used without sampling (i.e. using probit or Laplace bridge). Note that this kind of approximation is incompatible with the other methods you mentioned (i.e. ensembles, HMC, SWAG, and also standard VI) since these methods just yield samples/distributions over *weights* while our LA variants implicitly yield a Gaussian distribution over *logits* due to the linearization of the NN induced by the use of the GGN/Fisher (see [25] for details) or the use of only the last layer. While one could still apply linearization to other methods, this would not be theoretically justified, in contrast to GGN-/last-layer-LA. We are sorry about the confusion -- we will explain this more prominently. Further, the reason we benchmark our deterministic, probit-based version is that we found it to *consistently perform on par or better* than MC sampling. If we predict with the LA using MC samples on the logits, the runtime is only ~20% slower than the deterministic probit approximation which is still significantly faster than all other methods. In summary, we believe that the ability to obtain calibrated predictions *with a single forward-pass* is a critical and distinctive advantage of the LA over almost all other Bayesian DL / ensemble methods. We will emphasize this more in the paper.
>
> **OOD detection results .** You are right that the LA is mostly less confident (than the underlying MAP) which is an inherent property of the LA (allowing it to effectively mitigate overconfidence of the MAP) -- this is what MacKay calls [“moderated output”](https://authors.library.caltech.edu/13796/1/MACnc92d.pdf). We do not use any additional tuning that would lead to less confidence, this is simply the outcome of maximizing the marginal likelihood.
>
> **CSGHMC results.** In fact, all results labelled as *HMC* in the main paper refer to results for CSGHMC (i.e. we just used *HMC* as an abbreviation for CSGHMC; see lines 246-247). We will fix this. Sorry for the confusion.
>
> **Comparison between Hessian factorizations.** This is a great point, thank you for the suggestion! We conducted an experimental ablation for a full and dense Hessian approximation compared to the KFAC and diagonal variants on FashionMNIST (MNIST is typically too simple and all methods perform equally well) on a small convolutional network where it’s tractable. The table below reports the test log-likelihoods (LL, higher is better), test accuracies (Acc, higher is better) and AUROC scores  for OOD detection (averaged across all OOD datasets; higher is better) for one run with online marginal likelihood tuning and the full-network MC predictive. We also report the test inference time in seconds (averaged across the in-distribution and OOD datasets).
>
> | Method | LL     | Acc   | AUROC | Time (s)|
> |--------|--------|-------|-------|---------|
> | Diag   | -0.293 | 0.899 | 0.842 |  30.67  |
> | KFAC   | -0.276 | 0.903 | 0.836 |  35.22  |
> | Full   | -0.279 | 0.902 | 0.878 | 137.85  |
>
>
> These results show that the full/dense Hessian approximation is **significantly slower** than the diagonal and KFAC variants while not offering a tremendous performance benefit (only in terms of AUROC). We will add a more comprehensive evaluation on this to the paper.
>
> ## Discussion & Questions
> **Unsure whether it is a good fit for NeurIPS**. We address the concerns regarding novelty, significance and relevance in the general response. In short, we emphasize several further contributions in addition to the software package: the empirical evaluation, the introduction of the WILDS benchmark to the BDL community, the review of recent advances of the LA in deep learning, and the introduction of some new methods. Independently of that, we would like to emphasize that the [NeurIPS CfP](https://nips.cc/Conferences/2021/CallForPapers) explicitly invites papers on *Infrastructure (e.g., datasets, competitions, implementations, libraries)*, and that there indeed is a growing list of successful software packages published at NeurIPS and other top ML conferences (see the general response for a small selection of such papers).
>
> ## Software
> Thank you very much for taking the time to look into the package, this is very helpful! Of course, the proposed package is at the initial version and we will further improve and extend it in the future. Feedback like yours helps us to do so, and, in case of acceptance, the publicity would hopefully help us to identify further feature requests.
>
> **The code is oriented around research.**  Can you please clarify why is this the case? Are there particular things we could change to make it appeal more to practitioners? The library is certainly aimed at facilitating Bayesian DL methods in practical applications, in addition to enabling further research and developments.
>
> **`asdfghjkl`.** The `asdfghjkl` library is developed by Kazuki Osawa (*not* an author of this paper) and is now publicly available [here](https://github.com/kazukiosawa/asdfghjkl). He kindly provided us with early access to his library so we could include it as an additional backend besides BackPACK. The advantage of `asdfghjkl` is that it provides a larger variety of approximations and is less restrictive than BackPACK regarding the model architecture. For example, BackPACK does not work with residual connections, while `asdfghjkl` does. Further, `asdfghjkl` can be used in a distributed learning setup across multiple GPUs. Regarding convolutional layers supported, BackPACK currently supports [most commonly-used layers](https://docs.backpack.pt/en/master/supported-layers.html).
>
> **Too focused on KFAC.** The reason for this is that KFAC requires the most work to implement properly, which also explains its poor adoption rate. As shown in our paper, the KFAC variants provide the best performance at almost optimal scalability. For example, diagonal variants are typically not (much) more efficient but consistently perform worse and dense approximations are intractable in the deep learning setting.
>
> **Adding different Hessian approximations.** To add a different way of computing a dense Hessian (approximation), one only needs to inherit the `CurvatureInterface` class and implement the `def full(self, x, y, **kwargs)` method to return the log-likelihood Hessian. The new interface can then simply be passed to the Laplace.
>
> **Low-rank LA.** Adding other methods as you suggested (i.e. a low-rank Laplace using, for example, HVPs) would be possible but requires 1) adding backend support, which should be easy, and 2) another Laplace subclass dealing with low-rank quantities. This involves more effort but our framework makes clear which methods need to be implemented, and it is *far simpler* than implementing something similar from scratch. To the best of our knowledge, there is no prior work on scalable low-rank Laplace approximations, although that would be very interesting to add and investigate further independently. Are you aware of such work that we could provide within our software? One reason we did not implement anything based on HVPs is that such approaches typically scale poorly. For *one iteration* of the power method, we require *one pass through the data* which is only justifiable for small data sets or models.

---

> > ### Comment · Reviewer_vbUc · 2021-08-25
> > **Thanks**
> >
> > Thanks for the detailed rebuttal, I'm currently uninclined to change my score. Flagging that there has been discussion amongst the reviewers and AC; I'm mostly summarizing my thoughts after this discussion.
> >
> > A few comments:
> >
> > **Fig 5** I still maintain that it remains a bit of an unfair comparison. There are a) deterministic VI approaches that enable sampling free inference for classification [Wu et al, ICLR, '19] and b) it's straightforward to Taylor expand the logits with respect to the parameters and then use a weight space prior (e.g. the SWAG posterior or another Gaussian approximation) and then use probit / Laplace bridge. Alternatively, you could also have compared to the posterior mean in weight space from a CSGHMC approximation or the posterior mean of SWAG (which is the SWA solution). Both of which empirically perform pretty well: for a reference on the second, see Fig 2 of Maddox et al, '19 (amongst other papers).
> >
> > **CSGHMC results** Thank you for the clarification, please ensure that this is clearly marked in the final version, as this is pretty unsettling experimentally.
> >
> > **Full Hessian** Thanks for the clarification experiment. It's interesting to me that diagonal is pretty similar to KFAC here, I guess due to the shallowness of the network.
> >
> > **Software**
> >
> > Yes, by being research oriented, I mean that it generically seems to require the user to get pretty hands on to run experiments and to tune things. Thanks for getting the other library open-sourced, I'm interested in actually playing around with that one as well.
> >
> > Ultimately, my concern is that a full Hessian (which is what most practitioners would assume to be the first try for a Laplace approximation) is reasonably difficult to implement (a month or so after looking at your code the first time, I'm not sure now what other methods I'd need to implement).
> >
> > Poor scaling wrt the low-rank LA is a bit of a misnomer. From a research perspective, it's not particularly an issue to wait around for a few epochs to get a good low-rank approximation. It's also incorrect to state that power iteration requires one epoch per iteration when "stochastic power iteration" [Xu, et al, AISTATS, 18] exists and has been implemented for eigenvalue estimation of deep Hessians (https://github.com/noahgolmant/pytorch-hessian-eigenthings). Digging through the code of some other authors, one finds both Hessian vector product studies in PyTorch (https://github.com/amirgholami/PyHessian) and Fisher vector product studies (https://github.com/amzn/xfer/tree/master/finite_ntk). See publications linked in repo for both.
> >
> > **Conclusion** This refers back to the general rebuttal, but I think it should stay here. Overall, flipping through the software packages you listed, almost all of them provide novel capabilities of some sort - Espresso was first binary CNN, Jax MD was first MD simulation on the GPU w/ autograd, DeepOBS was a well regarded benchmarking suite, etc... - whereas I don't really see that from this software. KFAC in PyTorch has existed for a while, while Laplace approximation based papers have done similar evaluations to yours.

---

> > > ### Author Response · Authors · 2021-08-27
> > > **Thanks for your further comments!**
> > >
> > > Thank you for your further comments. We address all your concerns here. Nevertheless, could you please clarify what concrete changes we can make to the paper such that you would be inclined to raise your score and recommend accepting the paper?
> > >
> > > To address your main concern raised in your conclusion, we first want to refer to our general response, where we extensively highlight our contributions other than the library (such as establishing the new WILDS benchmark in the BDL community and proposing new Laplace variants). Moreover, while most of the individual components implemented in the library are indeed not novel, the consolidation of these methods enables new approaches, such as the continual learning experiment with online marginal likelihood optimization. We expect our library to open up the space of applications of the Laplace approximation, also to areas that were challenging to approach via Bayesian deep learning methods so far (e.g. we demonstrate in Section 4.3 that last-layer LA enables us to easily bring the benefits of Bayesian modeling to large-scale BERT-like transformer models for NLP tasks). In particular, we believe that the Laplace approximation is especially appealing for practical applications (compared to other Bayesian methods) due to 1) the ability to fit it _post-hoc_ on top of a MAP estimate, and 2) the fast inference time, requiring only a single forward pass for making predictions.
> > >
> > > As a side note, there are papers which propose a software package without any novel methods, e.g. [“Bayesian Layers” by Tran et al. (NeurIPS 2019)](https://proceedings.neurips.cc/paper/2019/file/154ff8944e6eac05d0675c95b5b8889d-Paper.pdf). They are nevertheless very useful for the community and their contributions should not be discounted.
> > >
> > >
> > > **Fig. 5**
> > >
> > > To clarify the intention behind Fig. 5: It is _not_ to show that Laplace is generally faster than all other methods, but to show that the default flavor of Laplace in our library is much faster than other existing Bayesian baselines, while being competitive in terms of uncertainty quantification (Fig. 4, 6 and Tab. 1). We will also clarify this directly in the paper.
> > >
> > > Nevertheless, a few comments regarding the specific methods you suggested:
> > >
> > > * The method of Wu et al. does not seem to be straightforwardly applicable to the large-scale convolutional architectures we considered (i.e. Wide-ResNet)---as far as we know, it has only been benchmarked for small networks on UCI data ([ref 1](https://arxiv.org/abs/2104.14421), [ref 2](https://arxiv.org/abs/1907.07504)).
> > > * While SWA indeed also just requires a single forward pass to make predictions, the figure you referred to shows that SWAG generally outperforms SWA. As we demonstrate that LA is competitive to SWAG, we would expect LA to generally outperform SWA.
> > > * While Taylor-expanding the logits for other methods (e.g. SWAG) is certainly possible, there is no evidence for the performance of such a predictive approximation. For LA with GGN, it has been shown that linearization leads to the correct predictive and therefore improves the performance (see [reference [25]](https://arxiv.org/pdf/2008.08400.pdf)).
> > > * We do not think that taking the mean of CSGHMC weight samples is conceptually sensible since they come from multiple posterior modes by design---the mean in weight space can lie in a low-probability region. Are you aware of work that shows otherwise?
> > >
> > > **Software**
> > >
> > > Our goal was to make the user interface as convenient as possible---could you please point out more concretely in what sense our library requires the user to get more hands-on than other machine learning libraries? The need to “tune things” is almost always present in machine learning, also/especially in production. Our library even provides automatic selection of some of these (e.g. prior precision optimization using marginal likelihood).
> > >
> > > In our response to your review we only briefly outlined what would be necessary for 1) full Hessian and 2) low-rank Hessian. Here, we would like to be more precise to highlight that such extensions are indeed simple -- in stark contrast to implementing such methods from scratch. We will provide a step-by-step guide in our library’s documentation and include the code in the next version of `laplace`.
> > >
> > > **Full Hessian**
> > >
> > > We just need to implement a backend that handles the full Hessian. We can use `ASDL` for this purpose and add the following to `laplace.curvature.asdl`:
> > >
> > > ```python
> > > from asdfghjkl.hessian import hessian_for_loss
> > >
> > > class AsdlHessian(AsdlInterface):
> > >     ...
> > >     def full(self, x, y, **kwargs):
> > >         hessian_for_loss(self.model, self.lossfunc, ‘full’, x, y)
> > >         H = self._model.hessian.data
> > >         loss = self.lossfunc(self.model(x).detach(), y)
> > >         return self.factor * loss, self.factor * H
> > >     ...
> > > ```
> > >
> > > Then, we simply need to pass the class above as the new backend. The main reason we did not implement this before is that it is not used in the context of neural networks because 1) it does not scale well and 2) it is not guaranteed to be positive semi-definite.
> > >
> > > **Low-Rank LA**
> > >
> > > While low-rank (+ diagonal) posterior approximations have been proposed in VI [using Bayes-by-Backprop](https://proceedings.neurips.cc/paper/2020/file/310cc7ca5a76a446f85c1a0d641ba96d-Paper.pdf), [noisy Adam variants](https://arxiv.org/pdf/1811.04504.pdf), or for SWAG, we are not aware of any work on low-rank LA. That being said, your idea of using HVPs together with the power method is interesting, so we tried it out. As already pointed out in our previous reply, we first need to extend the backend to provide us with `eigenvecs` and `eigenvals`. This will simply be a method of the `AsdlHessian` class above. `ASDL` implements the method `hessian_eigenvalues(model, lossfunc, data_loader, n_eigvals)` using the power method:
> > >
> > > ```python
> > > from asdfghjkl.hessian import hessian_eigenvalues
> > >
> > > # belongs to class `AsdlHessian`
> > > def eig_lowrank(self, data_loader):
> > >     eigvals, eigvecs = hessian_eigenvalues(self.model, self.lossfunc, data_loader,
> > >                                            top_n=self.low_rank, max_iters=self.low_rank*10)
> > >     eigvecs = torch.stack([torch.cat([p.flatten() for p in params]) for params in eigvecs], dim=1)
> > >     eigvals = torch.from_numpy(np.array(eigvals)).float()
> > >     loss = sum([self.lossfunc(self.model(x).detach(), y) for x, y in data_loader])
> > >     return eigvecs, self.factor * eigvals, self.factor * loss
> > > ```
> > >
> > > The second step is to add a `LowRankLaplace` class (see code snippet below). Using some linear algebra, we can efficiently compute the minimum necessary quantities `functional_variance`, `log_det_posterior_precision`, and `sample` (see [Mishkin et al. (2018)](https://arxiv.org/abs/1811.04504) Appendix C.4 for a derivation of the method used to efficiently implement `sample`). We find that the resulting method provides a good alternative to a full GGN with a rank of ~5% on a small example; see [this anonymous link](https://ibb.co/TmpkLP9) for a plot of the result. We are happy to include this in the software package and compare its performance with the other methods in the paper. We want to highlight again that this can be seen as a novel method and its implementation was very fast due to the entire backbone provided by our library. Thanks again for this suggestion.
> > >
> > > ```python
> > > class LowRankLaplace(BaseLaplace):
> > >     ...
> > >     @property
> > >     def V(self):
> > >         (U, l), prior_prec_diag = self.posterior_precision
> > >         return U / prior_prec_diag.reshape(-1, 1)
> > >
> > >     @property
> > >     def Kinv(self):
> > >         (U, l), _ = self.posterior_precision
> > >         return torch.inverse(torch.diag(1 / l) + U.T @ self.V)
> > >
> > >     def fit(self, train_loader):
> > >         # override fit since output of eighessian not additive across batch
> > >         eigenvectors, eigenvalues, loss = self.backend.eig_lowrank(train_loader)
> > >         self.H = (eigenvectors, eigenvalues)
> > >         self.loss = loss
> > >
> > >     @property
> > >     def posterior_precision(self):
> > >         self._check_fit()
> > >         return (self.H[0], self._H_factor * self.H[1]), self.prior_precision_diag
> > >
> > >     def functional_variance(self, Jacs):
> > >         prior_var = torch.einsum('ncp,nkp->nck', Jacs / self.prior_precision_diag, Jacs)
> > >         Jacs_V = torch.einsum('ncp,pl->ncl', Jacs, self.V)
> > >         info_gain = torch.einsum('ncl,nkl->nck', Jacs_V @ self.Kinv, Jacs_V)
> > >         return prior_var - info_gain
> > >
> > >     @property
> > >     def log_det_posterior_precision(self):
> > >         (U, l), prior_prec_diag = self.posterior_precision
> > > 	  return l.log().sum() + prior_prec_diag.log().sum() - torch.logdet(self.Kinv)
> > >
> > >     def sample(self, n_samples):
> > >         samples = torch.randn(self.n_params, n_samples)
> > >         d = self.prior_precision_diag
> > >         Vs = self.V * d.sqrt().reshape(-1, 1)
> > >         VtV = Vs.T @ Vs
> > >         Ik = torch.eye(len(VtV))
> > >         A = torch.cholesky(VtV)
> > >         B = torch.cholesky(VtV + Ik)
> > >         A_inv = torch.inverse(A)
> > >         C = torch.inverse(A_inv.T @ (B - Ik) @ A_inv)
> > >         Kern_inv = torch.inverse(torch.inverse(C) + Vs.T @ Vs)
> > >         dinv_sqrt = (1 / d).sqrt().reshape(-1, 1)
> > >         prior_samples = dinv_sqrt * samples
> > >         gain_samples = dinv_sqrt * Vs @ Kern_inv @ (Vs.T @ samples)
> > >         return self.mean + (prior_sample - gain_sample).T
> > > ```

---

> > > > ### Comment · Reviewer_vbUc · 2021-08-27
> > > > **Thanks for the detailed response**
> > > >
> > > > Thanks for the detailed response here. You're ultimately alleviating many of my concerns with the software based limitations of the package overall.
> > > >
> > > > With that being said, I think that my general comment about software packages also generally requiring methodological / experimental novelty still stands. We could go back and forth about this endlessly, but I do remember being most impressed about the Bayesian layers paper being the first to scale VI to massive language models (and to Bayesian MBRL).
> > > >
> > > > A couple of quick comments, mostly Fig 5:
> > > >
> > > > **Low Rank Laplace approximations**:
> > > > After a bit of digging, it seems like low rank Laplace approximations are reasonably popular in some specific corners of the bdl community, see for instance: Madras et al, ICLR, '20 (https://arxiv.org/abs/1910.09573), Maddox et al, '20 (https://arxiv.org/abs/2003.02139), Sharma et al, '21 (https://arxiv.org/abs/2102.12567).
> > > >
> > > > **Closed-Form Solutions**: If I recall correctly, Lu et al, 20 (https://arxiv.org/pdf/2006.07584.pdf, which you cite as [58]) use a closed form variant of the softmax + Gaussian approximation and use SWAG posteriors. Thus, it seems reasonable to compare to a closed form approximation.
> > > >
> > > > **CSGHMC**: Well, this is exactly the posterior mean, which should be useful as a summary.

---

> > > > > ### Author Response · Authors · 2021-08-30
> > > > > **Thanks for your engagement in this discussion!**
> > > > >
> > > > > We are glad that our responses resolved your technical concerns. Your only remaining concern appears to be on methodological/experimental novelty. We have addressed this thoroughly in our general response as well as in our conversation with you.
> > > > >
> > > > > Some quick final comments:
> > > > >
> > > > > - **Low-rank Laplace:** Thank you for the additional references, which we will add to the paper. The implementation we shared in our previous response enables these methods at scale with further capabilities, e.g., marginal-likelihood based hyperparameter optimization and an improved, linearized, predictive.
> > > > >
> > > > > - **Closed-form prediction:** We tried to use linearization+probit with our VB baseline. It yields a marginal improvement in uncertainty quantification (MNIST ECE 0.09; compared to 0.11 with sampling), but incurs a ~220% increase in computational cost and a marginal decrease in in-distribution accuracy (99.14% instead of 99.20%).
> > > > >
> > > > > - **CSGHMC:** We tried this and found that the CIFAR-10 (in-distribution) accuracy for the weight-space mean of CSGHMC is only 25%. This suggests that the posterior mean is unfortunately not a good summary of the CSGHMC samples.
> > > > >
> > > > > **To conclude the discussion, we would like to thank you once again for your review and continued engagement in this discussion, which helped us improve our paper.**

---

### Author Response · Authors · 2021-08-10
**General Response: Summary of Reviews and Clarification of our Contributions**

We would like to thank the reviewers for their time and efforts in providing detailed and insightful feedback, which we will incorporate in our revision.

We are pleased that the reviewers appreciated that our paper / `laplace` library “addresses an important problem” (Reviewer R8DH), is “well-timed [...] to remind the community of important progress in scalable LA” (Reviewer oDXG) and is “an important effort towards consolidation of scalable LA” (Reviewer oDXG). It was noted that, as a result, our work “should definitely be useful” (Reviewer vbUc), “could be of great interest to many researchers and practitioners in the field” (Reviewer R8DH) and “that it could be a boon to the community to see this type of demonstration published” (Reviewer vbUc).

We are also happy that the reviewers found that our `laplace` library is “pretty well put together” (Reviewer vbUc), “of reasonably high quality” (Reviewer vbUc), “well documented and [...] promises to be very user-friendly” (Reviewer T5ck), and that “overall it looks easy to tinker with the different choices/parameters” (Reviewer T5ck). Finally, we are glad that the reviewers highlighted that our “benchmarking [...] is no doubt diverse and reasonably exhaustive” (Reviewer oDXG) and that “the empirical results are [...] quite extensive” (Reviewer R8DH).

**We are thrilled that, as a result, the overall sentiment was positive, with three out of four reviewers recommending acceptance (Reviewers T5ck, oDXG, R8DH).**

Reviewer vbUc was hesitant to recommend acceptance “because almost the entire contribution is the software package.” While we certainly agree that our `laplace` library is a major contribution of our paper, **we would like to emphasize two important points for clarity**:
1. Firstly, **the [NeurIPS 2021 Call for Papers](https://neurips.cc/Conferences/2021/CallForPapers) explicitly invites "Infrastructure" papers that contribute libraries and implementations**, therefore appreciating the importance and relevance of such works to the NeurIPS community. While we agree that papers that include a software package as a main contribution are less common, we below provide a small selection of such works that were recently published at top ML conferences, for reference:
   - [Signatory (ICLR 2021)](https://openreview.net/forum?id=lqU2cs3Zca)
   - [Clairvoyance (ICLR 2021)](https://openreview.net/forum?id=xnC8YwKUE3k)
   - [JAX MD (NeurIPS 2020 **Spotlight**)](https://proceedings.neurips.cc//paper_files/paper/2020/hash/83d3d4b6c9579515e1679aca8cbc8033-Abstract.html)
   - [Neural Tangents (ICLR 2020 **Spotlight**)](https://openreview.net/forum?id=SklD9yrFPS)
   - [BackPACK (ICLR 2020 **Oral**)](https://openreview.net/forum?id=BJlrF24twB)
   - [Bayesian Layers (NeurIPS 2019)](https://papers.nips.cc/paper/2019/hash/154ff8944e6eac05d0675c95b5b8889d-Abstract.html)
   - [DeepOBS (ICLR 2019)](https://openreview.net/forum?id=rJg6ssC5Y7)
   - [Autoconj (NeurIPS 2018)](https://papers.nips.cc/paper/2018/hash/9b89bedda1fc8a2d88c448e361194f02-Abstract.html)
   - [Edward2 (NeurIPS 2018)](https://papers.nips.cc/paper/2018/hash/201e5bacd665709851b77148e225b332-Abstract.html)
   - [Espresso (ICLR 2018)](https://openreview.net/forum?id=Sk6fD5yCb)


2. Secondly, in our view, **our paper makes further significant contributions beyond the library**. We realize that we failed to communicate this clearly in the paper, and therefore highlight these additional contributions below to clarify this:
     1. **Empirical evaluation**: In our experiments, we comprehensively evaluate the different design choices of the LA to provide concrete practical recommendations and demonstrate that the LA has a **cost/performance trade-off that’s superior to many SotA baselines** (e.g. VI, CSGHMC, SWAG, deep ensembles). We cover a **diverse variety of downstream applications**, including out-of-distribution detection, distribution shift robustness, and continual learning. Notably, in Section 4.3 **we show results on [WILDS](https://wilds.stanford.edu)**, which is a recently-proposed, large-scale benchmark for realistic distribution shifts covering different data modalities and application domains. We there demonstrate the versatility of the LA by showing that last-layer-LA / `laplace` can easily be used to improve the calibration of large pre-trained models, including for **complex SOTA architectures such as BERT-like transformer models on NLP tasks**. This goes beyond the standard image classification setting into a regime where the application of other Bayesian deep learning methods is currently prohibitive (as a result, we only compare to non-Bayesian baselines there).
    2. **Establishing the new WILDS benchmark**: To our knowledge, we are the first to use [WILDS](https://wilds.stanford.edu) to benchmark distribution shift robustness of Bayesian deep learning (BDL) methods. Another contribution of our work is therefore **the initiative to establish the WILDS benchmark in the BDL community**. Leaders in the field have recently urged the community to start evaluating their methods on such realistic, practically-relevant benchmarks. For example, this was prominently discussed in the panel discussion of the recent [Uncertainty in Deep Learning workshop at ICML 2021](https://icml.cc/virtual/2021/workshop/8374) (see e.g. 04:34:00-04:40:00 and 04:50:30-04:55:30). Furthermore, the [abstract of the upcoming NeurIPS 2021 BDL workshop](https://nips.cc/Conferences/2021/Schedule?showEvent=21827) emphasizes that “the field of BDL itself is facing an evaluation crisis: most BDL papers evaluate uncertainty estimation quality of new methods on MNIST and CIFAR alone, ignoring needs of real world applications which use BDL. Therefore, [...] a particular focus of this year’s programme is on the reliability of BDL techniques in downstream tasks. […] We hope that the mainstream BDL community will adopt real world benchmarks based on such applications, pushing the field forward beyond MNIST and CIFAR evaluations.” **Our work therefore also aims to lead by example and contribute to overcoming the evaluation crisis in BDL.**
    3. **Review of the LA**: We provide a comprehensive **survey of recent advances** and present the components of scalable and practical Laplace approximations in deep learning, therefore providing a valuable reference for interested researchers and practitioners.
    4. **New methods / LA variants**: We in fact also provide some **new methods**. For example, the original KFAC-Laplace used a Fisher KFAC while `laplace` also enables a GGN-LA and empirical-Fisher-LA. The GGN KFAC is more exact but more expensive while the empirical-Fisher is typically cheaper than Fisher KFAC because it does not require sampling. Further, we propose _post-hoc_ marginal likelihood optimization in contrast to [Immer et al. 2021](https://arxiv.org/abs/2104.04975) who use it online only. There are plenty such examples, since we provide a much larger design space than previously considered methods by incorporating several Hessian approximations. We acknowledge that this methodological contribution might be viewed as somewhat incremental, but we nevertheless believe this to be useful.

**We believe that the combination of all of this makes for a strong overall contribution that is of high significance and relevance to the NeurIPS community** (as also recognized by the reviewers).

We further address the remaining concerns and questions of all reviewers in individual responses to their reviews.

---

### Decision · Program_Chairs · 2021-09-27

**Decision:**

Accept (Poster)

**Comment:**

A well-written paper proposing a software package.  The paper additionally gives a compact review on Laplace approximation (LA) for DNN  and benchmark experiments where LA works better or is comparable to more accurate approximation methods.  The software is easy to use but reviewers have concerns about its flexibility (e.g., prior is hardcoded and cannot be changed) and maintainability.  We hope that the authors would try to keep "Laplace" useful for the community.